# Multiplexed chemogenetics in astrocytes and motoneurons restore blood–spinal cord barrier in ALS

Najwa Ouali Alami[1,2,3,]*, Linyun Tang[1,]*, Diana Wiesner[1,4], Barbara Commisso[1], David Bayer[1,5], Jochen Weishaupt[1], Luc Dupuis[6], Phillip Wong[7,8], Bernd Baumann[9], Thomas Wirth[9], Tobias M Boeckers[4,10], Deniz Yilmazer-Hanke[3], Albert Ludolph[1,4], Francesco Roselli[1,4]

Blood–spinal cord barrier (BSCB) disruption is thought to contribute to motoneuron (MN) loss in amyotrophic lateral sclerosis (ALS). It is currently unclear whether impairment of the BSCB is the cause or consequence of MN dysfunction and whether its restoration may be directly beneficial. We revealed that $SOD1^{G93A}$, $FUS^{\Delta NLS}$, $TDP43^{G298S}$, and $Tbk1^{+/-}$ ALS mouse models commonly shared alterations in the BSCB, unrelated to motoneuron loss. We exploit PSAM/PSEM chemogenetics in $SOD1^{G93A}$ mice to demonstrate that the BSCB is rescued by increased MN firing, whereas inactivation worsens it. Moreover, we use DREADD chemogenetics, alone or in multiplexed form, to show that activation of Gi signaling in astrocytes restores BSCB integrity, independently of MN firing, with no effect on MN disease markers and dissociating them from BSCB disruption. We show that astrocytic levels of the BSCB stabilizers Wnt7a and Wnt5a are decreased in $SOD1^{G93A}$ mice and strongly enhanced by Gi signaling, although further decreased by MN inactivation. Thus, we demonstrate that BSCB impairment follows MN dysfunction in ALS pathogenesis but can be reversed by Gi-induced expression of astrocytic Wnt5a/7a.

## Introduction

Disruption of the blood–spinal cord barrier (BSCB) is a recently appreciated feature of amyotrophic lateral sclerosis (ALS), a disease affecting upper and lower motoneurons (MNs) with a progressive course and invariably fatal outcome (Hardiman et al, 2017). In spinal cord samples from ALS patients, BSCB impairment has been detected in the form of plasma protein leakage and reduced expression of endothelial tight junction (TJ) proteins (Garbuzova-Davis et al, 2012). Likewise, extravasation of erythrocytes,

immunoglobulins and plasma proteins; loss of endothelial TJ proteins (zonula occludens-1 [ZO-1], occludin, and claudin-5 [Zhong et al, 2008]); and decreased astrocytic end-feet and pericyte coverage (Garbuzova-Davis et al, 2007a, 2007b) have been detected in $SOD1^{G37R}$, $SOD1^{G85R}$, and $SOD1^{G93A}$ (Zhong et al, 2008; Winkler et al, 2014) as well as in $SOD1^{G93A}$ rats (Garbuzova-Davis et al, 2007a, 2007b) even at presymptomatic stages.

Nevertheless, the driver(s) of BSCB disruption in ALS and its ultimate impact on disease progression are still debated. It has been hypothesized that damage of the BSCB may arise as a cell autonomous consequence of mutant SOD1 accumulation in endothelial cells: endothelial cells isolated from $SOD1^{G93A}$ mice and immortalized mouse endothelial cells expressing the human $G93A$ mutant $SOD1$ display reduced levels of claudin-5 (CLN-5) and decreased transendothelial resistance (Meister et al, 2015).

Finally, the impact of the BSCB disruption on disease onset and progression is not univocal. It has been suggested that the extravasation of blood-derived factors, inflammatory cells, erythrocytes, and hemoglobin degradation products in itself is able to trigger the degeneration of MN. Indeed, administration of activated protein C (APC [Zhong et al, 2009]) or of iron chelators (Winkler et al, 2014) to $SOD1^{G93A}$ mice reduced the BSCB leakage and delayed disease progression. On the other hand, warfarin (an anticoagulant interfering with multiple vitamin K–dependent coagulation factors) aggravated BSCB disruption and worsened the degeneration of MN (Winkler et al, 2014). Nevertheless, none of these interventions are selective and may affect multiple disease cascades at once (e.g., reducing the transcription of the mutant $SOD1$ gene in the case of APC; Zhong et al, 2009).

Here, we have addressed the origin of the BSCB dysfunction in ALS and its weight on disease initiation and progression exploiting multiplexed chemogenetic strategies (involving both PSAM/PSEM[308] and DREADDs; Magnus et al, 2011; Roth, 2016) to enable interventions with high cell specificity and precise temporal control. We have

[1]Department of Neurology, Ulm University, Ulm, Germany   [2]International Graduate School in Molecular Medicine Ulm, Ulm, Germany   [3]Department of Neurology, Clinical Neuroanatomy, Ulm University, Ulm, Germany   [4]German Center for Neurodegenerative Diseases (DZNE), Ulm, Germany   [5]CEMMA Graduate School, Ulm University, Ulm, Germany   [6]Inserm U1118, Mécanismes Centraux et Périphériques de la Neurodégénérescence; Université de Strasbourg, Faculté de Médecine, Strasbourg, France   [7]Department of Pathology, Johns Hopkins University School of Medicine, Baltimore, MD, USA   [8]Department of Neuroscience, Johns Hopkins University School of Medicine, Baltimore, MD, USA   [9]Institute of Physiological Chemistry, Ulm University, Ulm, Germany   [10]Department of Anatomy and Cell Biology, Ulm University, Ulm, Germany

Correspondence: francesco.roselli@uni-ulm.de
*Najwa Ouali Alami and Linyun Tang contributed equally to this work

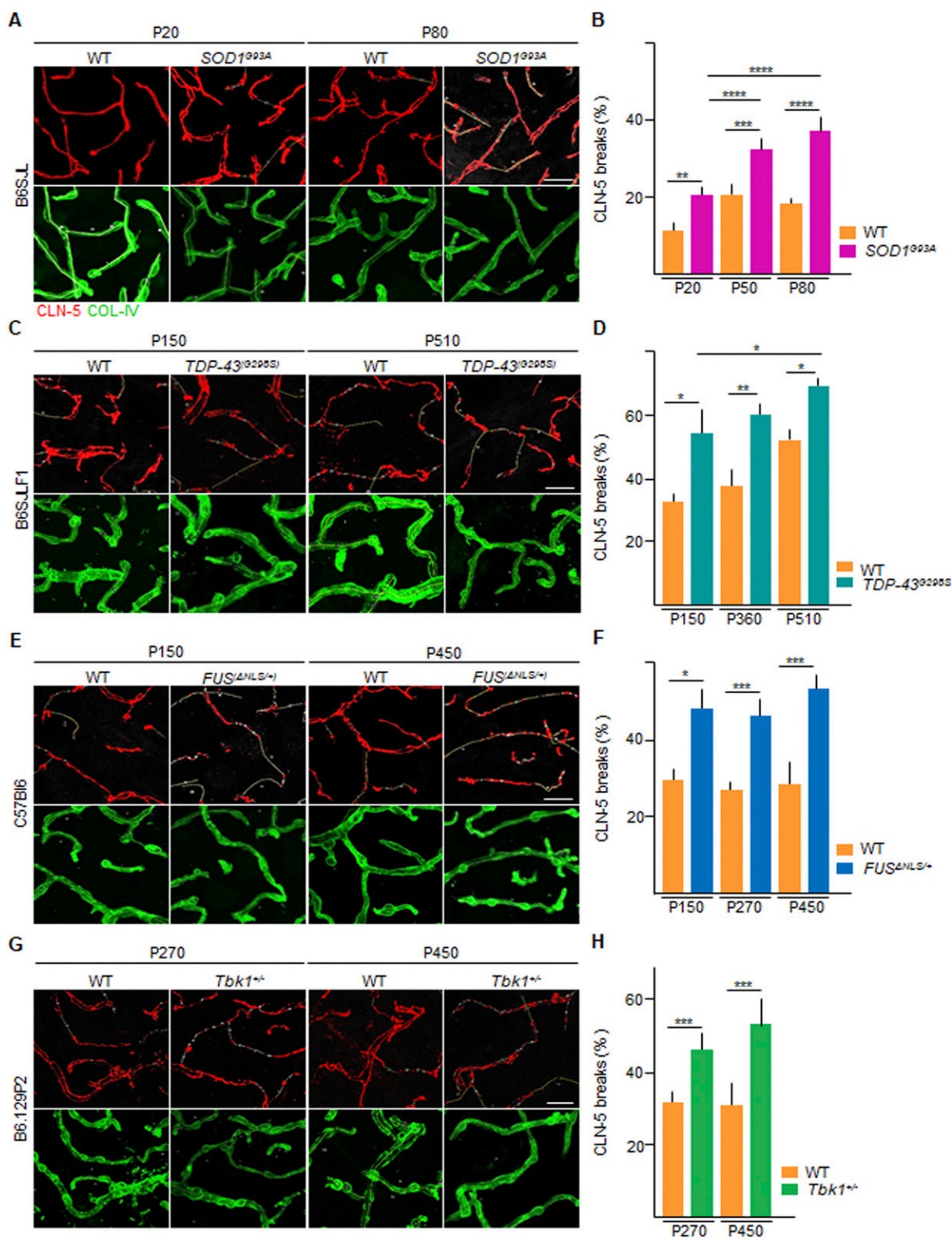

**Figure 1. Altered claudin-5 distribution in the spinal cord microvessels of *SOD1*, *FUS*, *TDP-43* and *Tbk1* amyotrophic lateral sclerosis mouse models.**
**(A, C, E, G)** Representative pictures of ventral horn of the lumbar spinal cord sections stained for claudin-5 (CLN-5 [red]) and collagen-IV (COL-IV [green]) of (A) P20 and P80 WT and *SOD1^{G93A}* mice (B6SJL background), (C) P150 and P510 WT and *TDP-43^{G298S}* mice (B6SJLF1 background), (E) P150 and P450 WT and *FUS^{ΔNLS/+}* mice (C57Bl6 background), and (G) P270 and P450 WT and *Tbk1^{+/−}* mice (B6.129P2 background). CLN-5 breaks are indicated by yellow lines (drawn with ImageJ software), underlining the discontinuity of CLN-5 ribbon. **(B, D, F, H)** Quantification of the capillary bed length (identified by COL-IV immunostaining) devoid of CLN-5 ribbon-like immunolabeling (break length) in the ventral horn of the spinal cord capillaries from (B) WT and transgenic *SOD1^{G93A}* mice (N = 4) at P20, P50, and P80; (D) WT and *TDP-43^{G298S}* mice (N = 3) at

demonstrated that BSCB disruption is a consequence of early, excitation-related MN dysfunction and that BSCB integrity can be restored (through the induction of Wnt proteins) by driving Gi signaling in astrocytes. Furthermore, we have used multiplexed chemogenetics to achieve a complete dissociation of BSCB integrity and disease burden, demonstrating how multiplexed chemogenetic can be used to untangle in vivo complex cellular interactions in BSCB disruption.

# Results

### Structural and functional disruption of the blood–spinal cord barrier is common to SOD1, FUS, TDP-43, and Tbk1 ALS mouse models

We set out to investigate the extent and progression of BSCB disruption in four murine ALS models characterized by distinct genetic mutations and different degrees of phenotype severity and progression rate. In particular, we considered the high-copy $SOD1^{G93A}$ line, together with the $FUS^{\Delta NLS/+}$ (Scekic-Zahirovic et al, 2017), the $TDP-43^{G298S}$ (Wiesner et al, 2018), and $Tbk1$ heterozygous knockout ($Tbk1^{+/-}$ [Brenner et al, 2019]) lines. In each ALS mouse strain, we investigated three time points at which critical pathological milestones were reached: (i) high-copy $SOD1^{G93A}$ mice were studied at P20 (appearance of ER stress and misfolded SOD1 buildup; Saxena et al, 2013), P50 (denervation of highly vulnerable fast fatigable MN; Pun et al, 2006), and P80 (appearance of overt neurological signs; Boillée et al, 2006; Ouali Alami et al, 2018); (ii) $FUS^{\Delta NLS/+}$ mice were studied at P150 (before appearance of neurological signs and denervation), P270 (appearance of denervation and clasping), and P450 (more advanced neurological signs; Scekic-Zahirovic et al, 2017); (iii) $TDP-43^{G298S}$ mice were studied at P150 (appearance of substantial neurological abnormalities; Wiesner et al, 2018), P360 (plateau of neurological dysfunction), and P510 (later stage in neurological dysfunction; subtle neurological abnormalities are present from the earliest time point but start worsening at around P150 with a plateau at around P250; Wiesner et al, 2018); and (iv) $Tbk1^{+/-}$ mice were used as a reference line for ALS disease at P270 and P450, when they showed lack of motor symptoms, weight loss, or premature death (Brenner et al, 2019).

First, we verified that the expression of the human mutant $SOD1$ transgene and the expression of the mutant $TDP-43$ transgene were persistently overexpressed and comparable across all time points (Fig S1A–D). Likewise, we verified that in knock-in $FUS^{\Delta NLS/+}$, the FUS protein was not overexpressed (as expected, Scekic-Zahirovic et al, 2016) but stayed stable over time (Fig S1E and F). Finally, TBK1 levels were steadily reduced in $Tbk1^{+/-}$ mice compared with WT mice (Fig S1G and H).

Next, we characterized the extent of MN loss over time in the four ALS mouse lines by quantifying the number of choline acetyl-transferase (ChAT)–positive cells in the ventral horn of the spinal cord. The $SOD1^{G93A}$ MN contingent was comparable with WT at P20 (average number of MN per lumbar spinal cord section: 21.2 ± 0.8 versus 22.4 ± 3.2, respectively) but steeply declined at P50 (15.7 ± 1.6 versus 22.8 ± 2.5 MN per section, respectively, $P < 0.001$), and was further aggravated at P80 (12.8 ± 2.3 versus 23.2 ± 2.7 MN per section, respectively; $P < 0.0001$; $SOD1^{G93A}$ P20 versus $SOD1^{G93A}$ P50: $P < 0.01$; $SOD1^{G93A}$ P20 versus $SOD1^{G93A}$ P80: $P < 0.0001$; Fig S2A and B; as previously reported: Ouali Alami et al, 2018). In the $TDP-43^{G298S}$ animals, a small loss of MN was detected already at P150 (19.6 ± 2.1 versus 26.0 ± 1.7 MN per section; $TDP-43^{G298S}$ versus WT; $P < 0.01$; Fig S2C and D) but did not significantly worsen at later time points (at P360: 17.4 ± 1.1 versus 25.3 ± 1.7 MN per section; $P < 0.001$ and at P510: 17.2 ± 1.6 versus 24.0 ± 3.1 MN per section; $P < 0.001$ $TDP-43^{G298S}$ versus WT, respectively; Fig S2C and D). In $FUS^{\Delta NLS/+}$ mice, the MN number was remarkably similar in mutant and WT animals at P150 (20.2 ± 2.2 versus 21.3 ± 1.5 MN per section, Fig S2E and F), but a significant MN loss appeared at P270 (16.3 ± 1.5 versus 22.0 ± 1.1 MN per section, respectively; $P < 0.001$; Fig S2E and F; as previously reported: Scekic-Zahirovic et al, 2017) which did not worsen further at P450 (14.0 ± 3.1 MN per section in $FUS^{\Delta NLS/+}$ versus 21.4 ± 1.1 MN per section in WT; $P < 0.0001$, $FUS^{\Delta NLS/+}$ P150 versus $FUS^{\Delta NLS/+}$ P270: $P < 0.1$; $FUS^{\Delta NLS/+}$ P150 versus $FUS^{\Delta NLS/+}$ P450: $P < 0.001$; Fig S2E and F). In the case of $Tbk1^{+/-}$ mice, the MN number was decreased at P270 compared with WT (16.8 ± 1.9 versus 20.3 ± 1.9 MN per section; $P < 0.1$; Fig S2G and H) but did not display a further decline at P450 (17.3 ± 1.6 versus 21.7 ± 2.1 MN per section, respectively; $P < 0.01$; Fig S2G and H). Thus, whereas the $SOD1^{G93A}$ displays a precipitous loss of MN, the $TDP-43^{G298S}$ and $Tbk1^{+/-}$ mice display a stable loss of a small contingent of MNs, whereas the $FUS^{\Delta NLS/+}$ mice show a late onset with limited MN loss.

We then proceeded to assess the extent of structural disruption of the BSCB in the four mouse lines, using as a readout to the extent of microvasculature, displaying focal loss of the TJ protein claudin-5 (CLN-5; Fig 1A–H) and the TJ protein ZO-1 (Fig 2A–H). In WT animals of each different mouse line, CLN-5 immunoreactivity was concentrated in the TJs between endothelial cells. CLN-5 protein forms a continuous ribbon inside the vessel walls (identified by the basal lamina component collagen-IV [COL-IV]; Fig 2A). Upon disruption of the BSCB, observed in concomitance with neuroinflammation or neurodegenerative processes (Ouali Alami et al, 2018), the ribbon of CLN-5 appears fragmented, forming "breaks" in the continuity of the CLN-5 ribbon. Consistent with previous reports (Ouali Alami et al, 2018), $SOD1^{G93A}$ mice displayed a significantly larger percentage of vessel length displaying CLN-5 breaks than WT littermates (20% ± 2% versus 12% ± 2%; $P < 0.01$) already at P20 (before any MN loss; Fig 1A and B). The extent of CLN-5 disruption of the BSCB further increased at P50 (33% ± 3% versus 21% ± 3%; $SOD1^{G93A}$ versus WT, respectively; $P < 0.001$) and at P80 (38% ± 4% versus 19% ± 1%; $SOD1^{G93A}$ versus WT, respectively, $P < 0.0001$; $SOD1^{G93A}$ P20 versus $SOD1^{G93A}$ P50: $P < 0.0001$; $SOD1^{G93A}$ P20 versus $SOD1^{G93A}$ P80: $P < 0.0001$; Fig 1A and B). In $TDP-43^{G298S}$ mice, CLN-5 distribution in the BSCB was already compromised at P150 (54.4% ± 8.1% versus 33.4% ±

P150, P360, and P510; (F) WT and $FUS^{\Delta NLS/+}$ mice (N = 3) at P150, P270, and P450; and WT and $Tbk1^{+/-}$ mice (N = 3) at P270 and P450. The quantifications are expressed as % of the total vessel length (n = 6 sections per mouse). Scale bars: 20 $\mu$m. Data information: in (B, D, F, H), data are presented as means ± SD. *$P < 0.01$, **$P < 0.001$, ***$P < 0.0001$, ****$P < 0.0001$ (two-way ANOVA [genotype × time point] with Bonferroni correction for multiple comparisons).

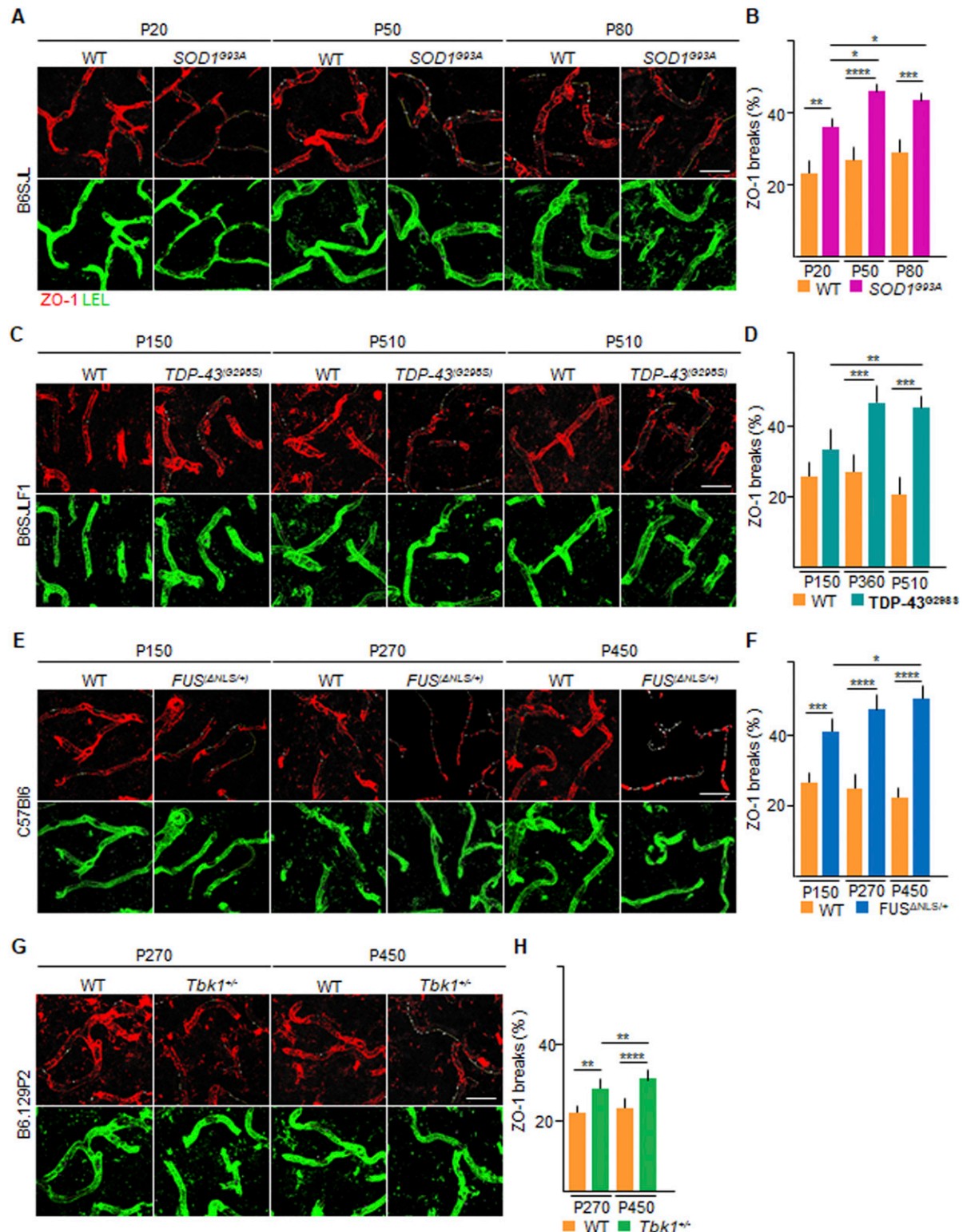

**Figure 2. Altered zonula occludens-1 (ZO-1) distribution in the spinal microvessels of *SOD1*, *FUS*, *TDP-43* and *Tbk1* amyotrophic lateral sclerosis mouse models.**
**(A, C, E, G)** Representative pictures of ventral horn of the lumbar spinal cord sections stained for ZO-1 (red) and *Lycopersicon esculentum* (tomato) lectin (LEL [green]) of (A) P20, P50, and P80 WT and *SOD1*^G93A^ mice (B6SJL background); (C) P150, P360, and P510 WT and *TDP-43*^G298S^ mice (B6SJLF1 background); (E) P150, P270, and P450 WT and *FUS*^ΔNLS/+^ mice (C57Bl6 background); and (G) P270 and P450 WT and *Tbk1*^+/−^ mice (B6.129P2 background). ZO-1 breaks are indicated by yellow lines (drawn with ImageJ software), underlining the discontinuity of ZO-1 ribbon. **(B, D, F, H)** Quantification of the capillary bed length (identified by LEL immunostaining) devoid of ZO-1 ribbon-like immunolabeling (break length) in the ventral horn of the spinal cord capillaries from (B) WT and transgenic *SOD1*^G93A^ mice (N = 4) at P20, P50, and P80; (D) WT and *TDP-43*^G298S^ mice (N = 3) at P150, P360, and P510; (F) WT and *FUS*^ΔNLS/+^ mice (N = 3) at P150, P270, and P450; and WT and *Tbk1*^+/−^ mice (N = 3) at P270 and P450. The quantifications are expressed as % of the total vessel length (n = 6 sections per mouse). Scale bars: 20 μm. Data information: in (B, D, F, H), data are presented as means ± SD. *P < 0.01, **P < 0.001, ***P < 0.0001, ****P < 0.0001 (two-way ANOVA [genotype × time point] with Bonferroni correction for multiple comparisons).

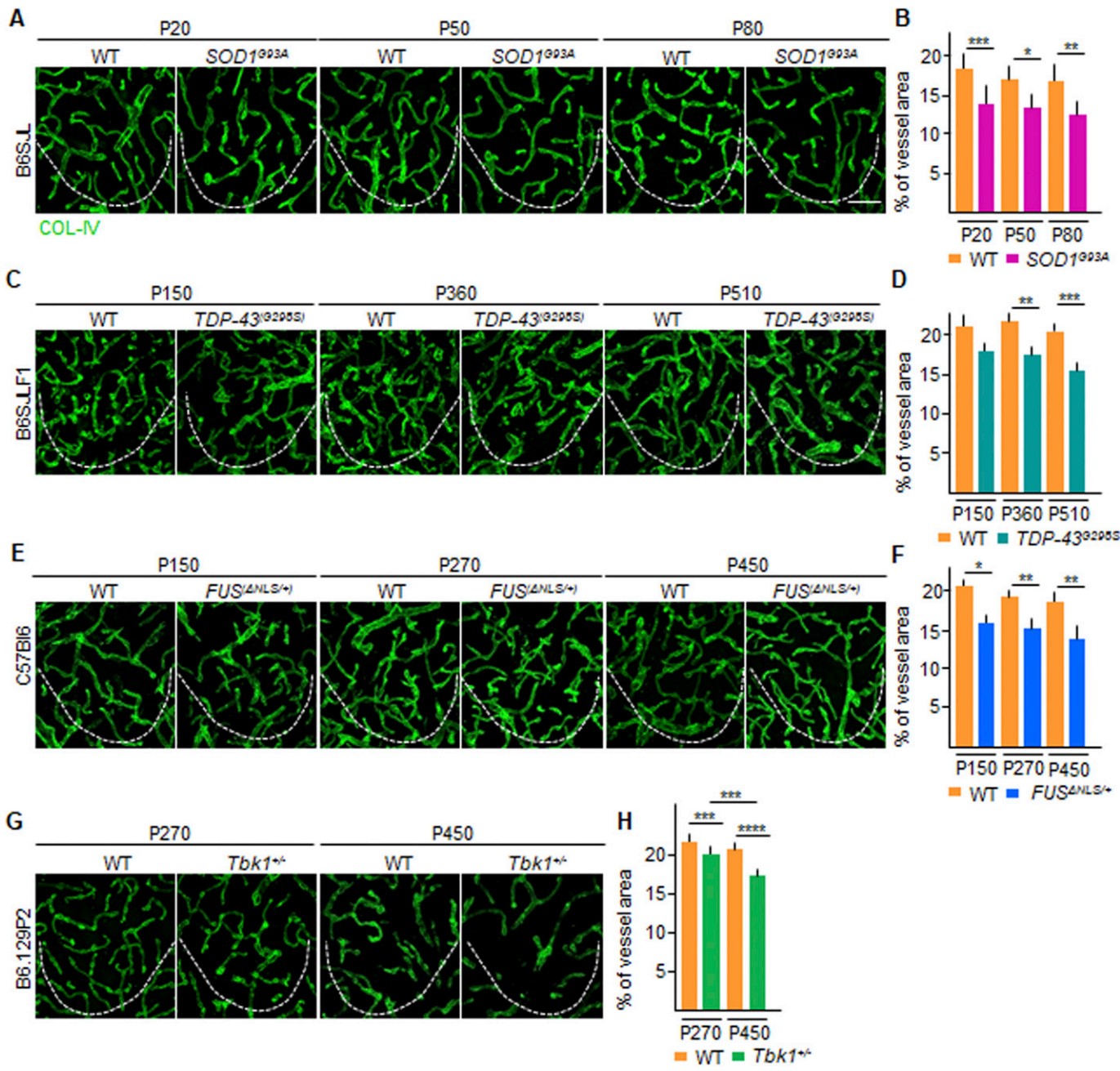

**Figure 3. Overall vascular density in *SOD1*, *FUS*, *TDP-43* and *Tbk1* amyotrophic lateral sclerosis mouse models.**
**(A, C, E, G)** Representative pictures of microvasculature ramification in the ventral horn of the lumbar spinal cord of (A) P20, P50, and P80 WT and *SOD1^G93A* mice (B6SJL background); (C) P150, P360, and P510 WT and *TDP-43^G298S* mice (B6SJLF1 background), (E) P150, P270, and P450 WT and *FUS^ΔNLS/+*mice (C57Bl6 background); and (G) P270 and P450 WT and *Tbk1^+/−* mice (B6.129P2 background). The microvessel walls are stained with collagen-IV (COL-IV [green]), specific for type 4 collagen in the basal lamina. **(B, D, F, H)** Quantification of the COL-IV+ area in the ventral horn of the spinal cord from (B) WT and transgenic *SOD1^G93A* mice (N = 4) at P20, P50, and P80; (D) WT and *TDP-43^G298S* mice (N = 3) at P150, P360, and P510; (F) WT and *FUS^ΔNLS/+* mice (N = 3) at P150, P270, and P450; and WT and *Tbk1^+/−* mice (N = 3) at P270 and P450. The quantifications are expressed as % of COL-IV+ area per total area (n = 6 sections per mouse). Scale bars: 10 μm. Data information: in (B, D, F, H), data are presented as means ± SD. *$P < 0.01$, **$P < 0.001$, ***$P < 0.0001$, ****$P < 0.0001$ (two-way ANOVA [genotype × time point] with Bonferroni correction for multiple comparisons).

3.2%; *TDP-43^(G298S)* versus WT; $P < 0.1$; Fig 1C and D) and had progressively worsened at P360 (60.2% ± 5.4% versus 37.1% ± 8.1%, respectively; $P < 0.01$; Fig 1C and D) and at P510 (70.5% ±1.9% versus 53.4% ± 3.4%, in *TDP-43^(G298S)* versus WT, respectively; $P < 0.1$; *TDP-43^(G298S)* P150 versus *TDP-43^(G298S)* P510: $P < 0.1$; Fig 1C and D), despite

no further loss of MN. In *FUS^ΔNLS/+* mice, breakdown of the CLN-5 ribbon was detected already at P150 (before signs of MN loss or of neurological impairment; 48.4% ± 7.5% versus 30.5% ± 4.1% in *FUS^ΔNLS/+* versus WT mice; $P < 0.1$; Fig 1E and F) and became more intense at P270 (47.6% ± 6% versus 24.5% ± 2.1%; $P < 0.001$) and at

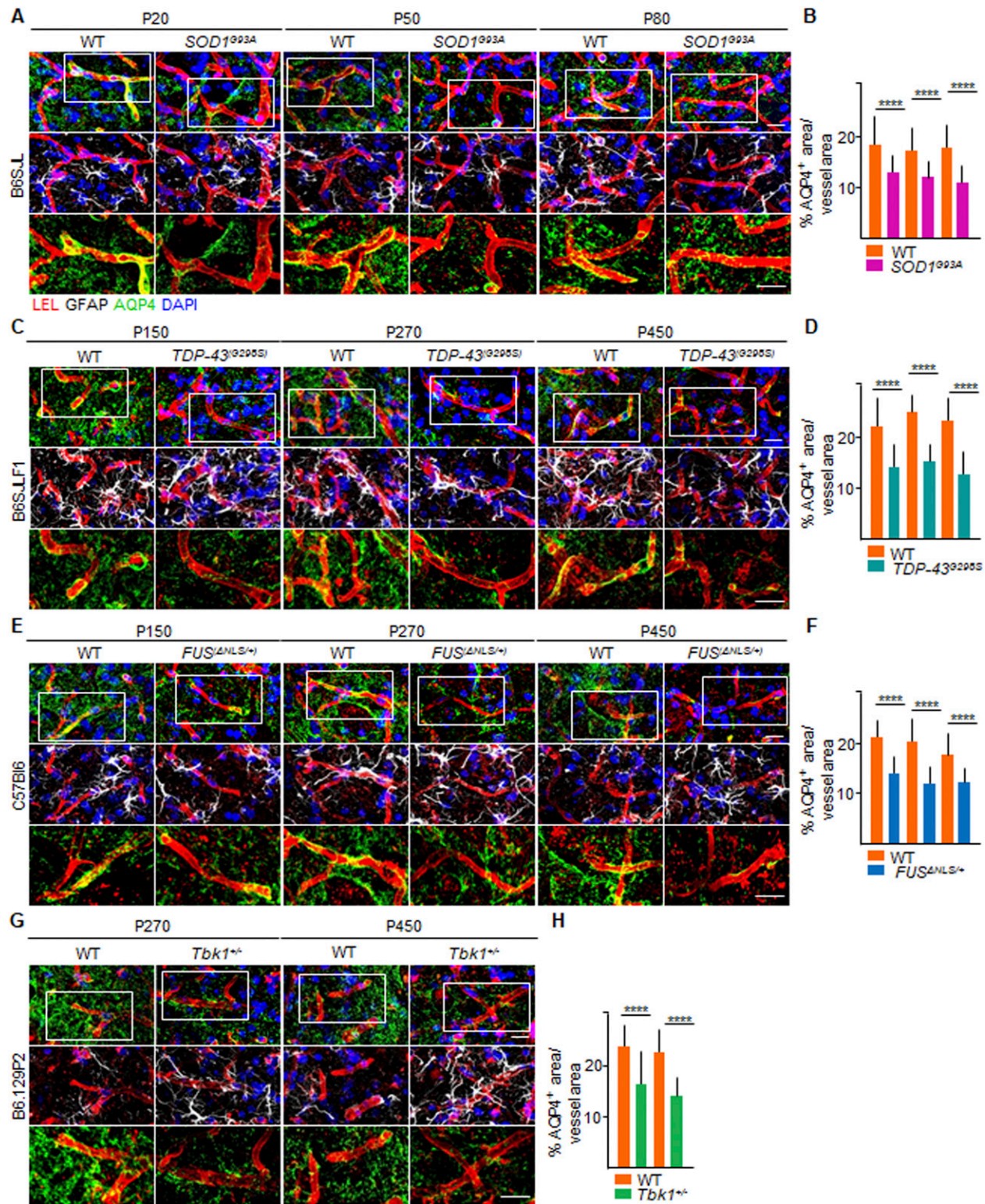

**Figure 4. Altered aquaporin-4⁺ astrocytic end-feet coverage of spinal cord vessels in *SOD1*, *FUS*, *TDP-43* and *Tbk1* amyotrophic lateral sclerosis mouse models.**
**(A, C, E, G)** Representative images of astrocytic end-feet coverage around the vessels (red), immunostained with aquaporin-4 (AQP4 [green]) in the ventral horn of the lumbar spinal cord of (A) P20, P50, and P80 WT and *SOD1^G93A^* mice (B6SJL background); (C) P150, P360, and P510 WT and *TDP-43^G298S^* mice (B6SJLF1 background); (E) P150, P270, and P450 WT and *FUS^ΔNLS/+^* mice (C57Bl6 background); and (G) P270 and P450 WT and *Tbk1^+/-^* mice (B6.129P2 background). The first row shows the overlap of AQP4 surrounding the vessels, labeled with *Lycopersicon esculentum* (tomato) lectin (LEL [red]); the second row shows the extremity of astrocytes, immunolabeled with glial fibrillary acidic protein (white), wrapping the capillaries (red). Nuclei are detected by DAPI staining (blue); the third row represents a high magnification of the insert in the

P450 (52.1% ± 4.4% versus 29.2% ± 9.1% in $FUS^{\Delta NLS/+}$ versus WT littermates; $P$ < 0.001; Fig 1E and F). Most notably, we detected a significant disruption of CLN-5 ribbon also in $Tbk1^{+/-}$ mice at P270 (46.1% ± 6.2% versus 32.6 ± 5.1; in $Tbk1^{+/-}$ versus WT; $P$ < 0.001; Fig 1G and H) with a further increase finally observed at P450 (53.2% ± 9.3% versus 29.0% ± 9.1% in $Tbk1^{+/-}$ and WT, respectively; $P$ < 0.001; Fig 1G and H), although these mice did not display overt signs of disease (Brenner et al, 2019).

To demonstrate that the appearance of CLN-5 ribbon breaks corresponds to a comprehensive disruption of the BSCB organization, we explored the distribution and localization of ZO-1 (Fig 2A–H), an intracellular TJ protein largely unrelated to claudins (Vogelmann & Nelson, 2005). In WT vessels, ZO-1 immunolocalization was similar to CLN-5, displaying a continuous ribbon inside the COL-IV boundary. In the $SOD1^{G93A}$ mice, at P20, the percentage of vessel length displaying breaks of ZO-1 ribbons was higher than that in the WT littermates (36.3% ± 2.1% versus 23.8% ± 4.5%, respectively; $P$ < 0.01; Fig 2A and B). At P50, the capillary disruption is reflected with a further increase in the percentage of ZO-1 breaks (46.0% ± 1.7% of breaks in the $SOD1^{G93A}$ compared with 26.5% ± 4.6% in the WT littermate; $P$ < 0.0001; Fig 2A and B). This increment has also been observed at P80 (44.1% ± 2.9% versus 29.2% ± 2.7% in $SOD1^{G93A}$ versus WT, respectively, $P$ < 0.001; $SOD1^{G93A}$ P20 versus $SOD1^{G93A}$ P50: $P$ < 0.1; $SOD1^{G93A}$ P20 versus $SOD1^{G93A}$ P80: $P$ < 0.1; Fig 2A and B). In the $TDP-43^{G298S}$ animals, ZO-1 fragmentation was slightly higher at P150 than WT littermates (32.4% ± 4.9% versus 25.5% ± 4.1%, respectively; Fig 2C and D), and significantly increased at P360 (47.6% ± 4.5% versus 27.5% ± 5.1%, respectively; $P$ < 0.001; Fig 2C and D) and at P510 (45.1% ± 3.4% versus 20.1% ± 5.1%, respectively; $P$ < 0.001; $TDP-43^{(G298S)}$ P150 versus $TDP-43^{(G298S)}$ P360: $P$ < 0.01; Fig 2C and D). In $FUS^{\Delta NLS/+}$ mice, ZO-1 distribution was already significantly compromised at P150 (40.3% ± 4.1% versus 25.8% ± 2.1% $FUS^{\Delta NLS/+}$ versus WT, $P$ < 0.001; Fig 2E and F) and further increased at P270 (46.8% ± 5.7% versus 25.6% ± 4.1% in $FUS^{\Delta NLS/+}$ versus WT; $P$ < 0.0001, Fig 2E and F). The extent of ZO-1 ribbon fragmentation further worsened at P450 in $FUS^{\Delta NLS/+}$ mice (50.3% ± 4.3% versus 23.6% ± 2.4% in $FUS^{\Delta NLS/+}$ versus WT; $P$ < 0.0001; $FUS^{\Delta NLS/+}$ P150 versus $FUS^{\Delta NLS/+}$ P450: $P$ < 01; Fig 2E and F). Remarkably, we also detected loss of ZO-1 in the $Tbk1^{+/-}$ ALS mouse model at P270 (27.0% ± 0.6% versus 22.7% ± 0.7%; $Tbk1^{+/-}$ versus WT; $P$ < 0.01; Fig 2G and H), with a significant increase at P450 (31.4% ± 2.2% versus 23.5% ± 1.6%, respectively, in $Tbk1^{+/-}$ versus WT; $P$ < 0.0001; $Tbk1^{+/-}$ P270 versus $Tbk1^{+/-}$ P450: $P$ < 0.01; Fig 2G and H). Overall, these data suggest that fragmentation of TJ protein complexes is crucial in the maintenance of BSCB integrity and occurred before and independently of MN loss in the ALS mice considered.

We wondered whether the loss of TJ proteins may be related to an ongoing degeneration of the spinal cord microvascular bed. We assessed the density of microvessels in the ventral horn of spinal cord samples derived from the $SOD1^{G93A}$, $FUS^{\Delta NLS/+}$, $TDP43^{(G298S)}$,

and $TBK1^{+/-}$ mice at different time points and compared them with age- and strain-matched controls. At their baseline, all ALS mice already displayed a degree (~20% across the different lines) of reduced COL-IV+ microvessel density in the ventral horn of the spinal cord. More precisely, at P20 in the $SOD1^{G93A}$, the percentage of the COL-IV+ area in the region of interest (ROI) was decreased (14.2% ± 1.4%) compared with WT littermates (18.4% ± 1.3%; $SOD1^{G93A}$ versus WT; $P$ < 0.001; Fig 3A and B), which persisted unmodified at P50 (14.0% ± 2.0% versus 17.4% ± 1.2% in $SOD1^{G93A}$ versus WT mice, respectively; $P$ < 0.1; Fig 3A and B) with a slight decrease at P80 (12.6% ± 2.4% versus 17.0% ± 1.6%; $P$ < 0.01; Fig 3A and B). A similarly stable decrease in vascular density was also detected in $TDP-43^{G298S}$ mice at P150 (17.6% ± 1.5% versus 20.8% ± 2.2% in $TDP-43^{G298S}$ versus WT; Fig 3C and D), followed by a significant decrease at P360 (17.3% ± 1.1% versus 21.1% ± 1.4%, respectively; $P$ < 0.01; Fig 3C and D) and later at P510 (15.6% ± 1.1% versus 20.7% ± 1.3% in $TDP-43^{G298S}$ versus WT, respectively; $P$ < 0.001; Fig 3C and D). Notably, a significant decrease in vascular density was also detected in $FUS^{\Delta NLS/+}$ mice at P150 (16.6% ± 1.1% versus 20.6% ± 0.6% in $FUS^{\Delta NLS/+}$ versus WT littermates; $P$ < 0.1; Fig 3E and F), at P270 (15.6% ± 1.5% versus 19.2% ± 1.6%; $P$ < 0.01; Fig 3E and F), and at P450 (14.6% ± 2.3% versus 19.1% ± 1.5%; $P$ < 0.01; Fig 3E and F). Despite the absence of neurological abnormalities and MN loss, $Tbk1^{+/-}$ mice also displayed alteration in the vascular density (Fig 3G and H), with a significant reduction in the COL-IV+ area at P270 and at P450 (20.0% ± 1.1% and 17.0% ± 1.8%, respectively; $P$ < 0.001), in contrast to WT littermates (23.0% ± 1.1% and 21.1% ± 1.6%, respectively; $Tbk1^{+/-}$ P270 versus WT P270 $P$ < 0.001; $Tbk1^{+/-}$ P450 versus WT P450 $P$ < 0.0001; Fig 3G and H).

We further explored the possibility that TJ loss may be due to the focal degeneration of endothelial cells (in the so-called string vessel formation, defined as a collapsed basement membrane without the endothelium and with no function in circulation; Forsberg et al, 2018). We immunostained spinal cord sections for COL-IV+ together with CD31/PECAM-1, a membrane and cytoplasmic protein abundantly and constitutively expressed on endothelial cells (Feng et al, 2004; Caligiuri, 2019). Across all ALS lines and all time points, we observed an almost complete (>90%) coverage of COL-IV+ with CD31/PECAM-1+ immunostaining, confirming that every COL-IV+ vessel was indeed lined with endothelial cells (Fig S3).

Moreover, we assessed a third readout of BSCB integrity, namely, the extent of astrocytic end-feet vessel coverage (Kubotera et al, 2019). To this end, we immunostained spinal cord sections from the four ALS lines with aquaporin-4 (AQP4), a water channel protein, highly expressed in astrocytic end-feet (Nielsen et al, 1997), and glial fibrillary acidic protein (GFAP), a class-III intermediate filament and cell-specific marker that distinguishes astrocytes from other glial cells, combined with COL-IV. Whereas spinal cord vessels in WT mice (of any strain) displayed substantial coverage with AQP4+ processes, ALS mice exhibit a relevant decrease in AQP4+ coverage. Indeed $SOD1^{G93A}$ mice at P20 displayed a % of AQP4+ area equal to

first row (white rectangle) and displays the distribution of AQP4 on the LEL+ endothelium, as a marker of blood–spinal cord barrier stability. The magnified pictures do not include DAPI staining. Scale bar: 20 $\mu m$. **(B, D, F, H)** Quantification of the AQP4+ area, expressed as % of the total vessel area in the ventral horn of the spinal cord from (B) WT and transgenic $SOD1^{G93A}$ mice (N = 4) at P20, P50, and P80; (D) WT and $TDP-43^{G298S}$ mice (N = 3) at P150, P360, and P510; (F) WT and $FUS^{\Delta NLS/+}$ mice (N = 3) at P150, P270, and P450; and WT and $Tbk1^{+/-}$ mice (N = 3) at P270 and P450; (n = 6 sections per mouse). Scale bars: 10 $\mu m$. Data information: in (B, D, F, H), data are presented as means ± SD. ****$P$ < 0.0001 (two-way ANOVA [genotype × time point] with Bonferroni correction for multiple comparisons).

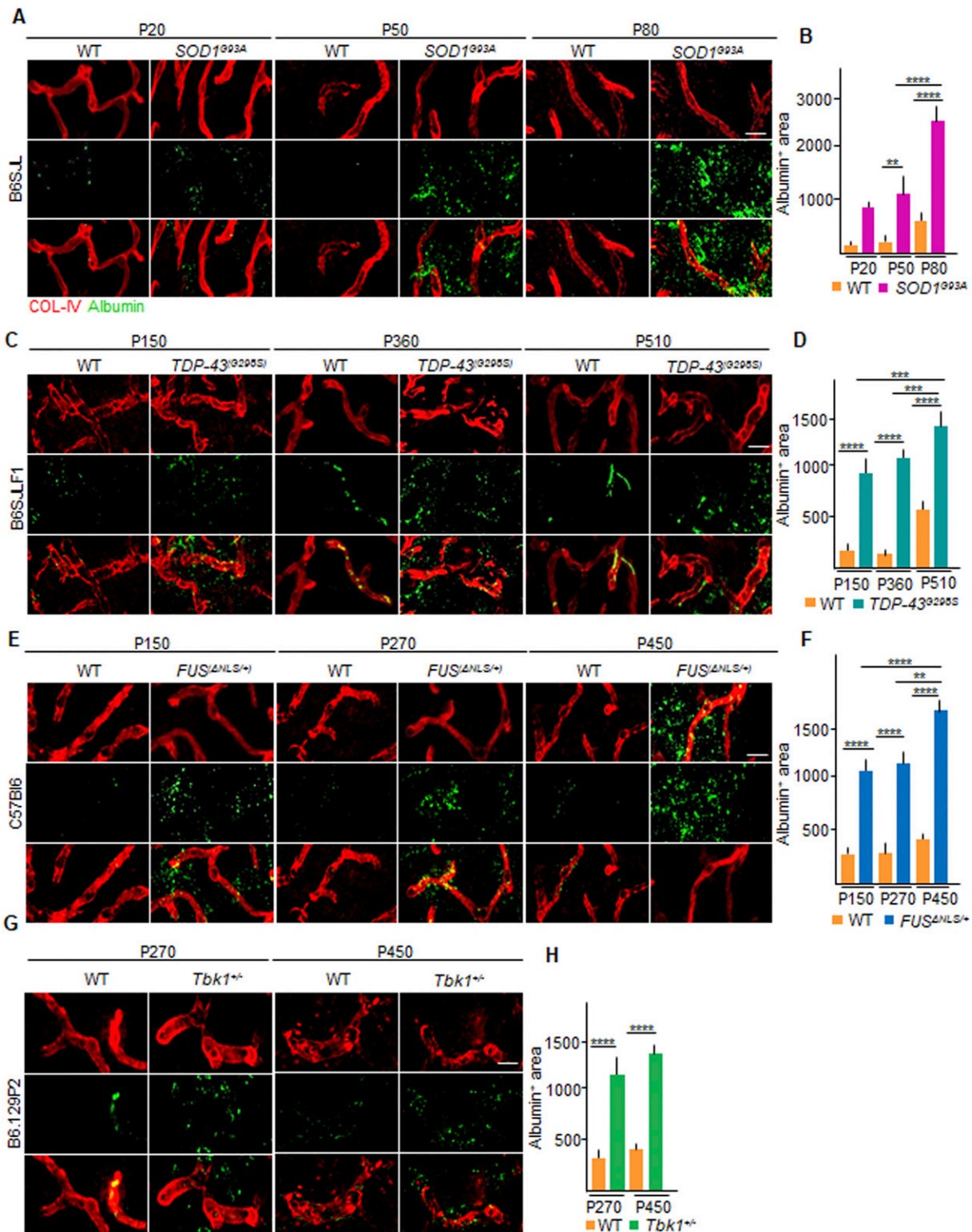

**Figure 5.  Albumin extravasation demonstrates functional impairment of the blood–spinal cordbarrier in SOD1, FUS, TDP-43, and Tbk1 amyotrophic lateral sclerosis mouse models.**

(A, C, E, G) Representative images displaying intraparenchymal albumin staining (green) around the vessels (COL-IV; red) localized in the ventral horn of the lumbar spinal cord of (A) P20, P50, and P80 WT and $SOD1^{G93A}$ mice (B6SJL background); (C) P150, P360, and P510 WT and $TDP-43^{G298S}$ mice (B6SJLF1 background); (E) P150, P270, and P450 WT and $FUS^{\Delta NLS/+}$ mice (C57Bl6 background); and (G) P270 and P450 WT and $Tbk1^{+/−}$ mice (B6.129P2 background). The first row shows vessels immunostained with COL-IV (red) alone. The second row shows the distribution of albumin (green) in the parenchyma in the ventral horn of the spinal cord. The third row represents images merged

13.20% ± 3.8% of the total vessel area, compared with 18.8% ± 4.6% of the total vessel area in corresponding WT littermates (P < 0.0001; Fig 4A and B). AQP4 coverage stayed stable either at P50 (12.1% ± 3.9% versus 17.8% ± 4.2% in SOD1$^{G93A}$ versus WT; P < 0.0001, Fig 4A and B) or at P80 (11.3% ± 4.1% versus 18.3% ± 4.5% of the total vessel area, respectively; P < 0.0001; Fig 4A and B). In TDP-43$^{G298S}$ mice, the AQP4$^+$ area was already significantly decreased at P150 (14.0% ± 4.3% of the vessel area) compared with WT littermates (22.1% ± 4.2% of the total vessel area; P < 0.0001; Fig 4C and D), despite no further decrease over time point P360 (15.0% ± 2.8% versus 24.1% ± 3.7% of the total vessel area, in TDP-43$^{G298S}$ mice versus WT, respectively; P < 0.0001; Fig 4C and D) and P510 (12.6% ± 3.9% versus 22.6% ± 5.0% of the total vessel area in TDP-43$^{G298S}$ mice versus WT littermates; P < 0.0001; Fig 4C and D). In FUS$^{ΔNLS/+}$ mice at P150, we also witnessed a conspicuous reduction in the astrocytic water channel AQP4 surrounding the blood microvessels (14.3% ± 3.6% of the total vessel area), in contrast to WT mates (21.0% ± 3.7%; P < 0.0001; Fig 4E and F). The amount of AQP4 lining FUS$^{ΔNLS/+}$ vessels did not diminish over time point P270 (12.1% ± 3.5% versus 20.6% ± 4.5%; FUS$^{ΔNLS/+}$ versus WT, respectively; P < 0.0001; Fig 4E and F) and P450 (12.6% ± 2.5% versus 19.7% ± 4.7%; FUS$^{ΔNLS/+}$ versus WT, respectively; P < 0.0001; Fig 4E and F). The same was observed in Tbk1$^{+/−}$ mice at P270 (Tbk1$^{+/−}$ versus WT, P < 0.0001) and P450 (Tbk1$^{+/−}$ versus WT, P < 0.0001; Fig 4G and H).

This effect cannot be attributed to a loss of astrocytes because an actual increase in the cell number (astrocyte hyperplasia; Fig S4B, E, H, and K) and size (hypertrophy, as measured by cumulative GFAP$^+$ area; Fig S4C, F, I, and L) was detected in the four ALS lines, either progressively increasing in the SOD1$^{G93A}$ mouse (i.e., P20: 12.4 ± 0.6 versus 9.6 ± 0.3 GFAP$^+$ cell/10$^4$ µm in SOD1$^{G93A}$ versus WT; P < 0.1; P50: 14.5 ± 1.1 versus 10.1 ± 0.4 GFAP$^+$ cell/10$^4$ µm, respectively; P < 0.001; P80: 15.3 ± 1.9 versus 11.3 ± 0.4 GFAP$^+$ cell/10$^4$ µm, respectively; P < 0.01; SOD1$^{G93A}$ P20 versus SOD1$^{G93A}$ P80: P < 0.1; Fig S4A and B), as previously reported (Ouali Alami et al, 2018) or stably increasing over the time points investigated for the TDP43 (Fig S4D and E), FUS (Fig S4G and H), and Tbk1 (Fig S4J and K) mice. The loss or retraction of AQP4$^+$ processes thus appears to be an additional aspect of the BSCB dysfunction.

We completed our evaluation of the BSCB integrity in the ALS mouse lines by assessing the functional impairment of the BSCB by quantifying the amount of extravasated intraparenchymal albumin in the different mouse models. WT animals always displayed limited parenchymal albumin immunoreactivity. On the other hand, in the SOD1$^{G93A}$ mice, albumin extravasation was already detectable at P20 (mean immunostaining intensity over the parenchymal area: 862 ± 54 versus 173 ± 60 a.u. in SOD1$^{G93A}$ versus WT; Fig 5A and B) and significantly increased at P50 (1,142 ± 492 versus 240 ± 50 a.u. in SOD1$^{G93A}$ versus WT; P < 0.01; Fig 5A and B) and at P80 (2,466 ± 394 versus 547 ± 287 a.u. SOD1$^{G93A}$ versus WT, respectively, P < 0.0001; SOD1$^{G93A}$ P50 versus SOD1$^{G93A}$ P80: P < 0.0001; Fig 5A and B). Likewise, in the TDP-43$^{G298S}$ mice, albumin extravasation was detectable at P150 (914 ± 202 versus 199 ± 103 a.u. in TDP-43$^{G298S}$ versus WT; P <

0.000; Fig 5C and D), with further increase detected at P360 (1,090 ± 83 versus 180 ± 43 a.u., respectively, in TDP-43$^{G298S}$ versus WT; P < 0.000; Fig 5C and D), along with augmentation at P510 (1,428 ± 212 a.u. in TDP-43$^{G298S}$ versus 568 ± 108 a.u. in WT; P < 0.0001; TDP-43$^{G298S}$ P150 versus TDP-43$^{G298S}$ P510: P < 0.001; TDP-43$^{G298S}$ P360 versus TDP-43$^{G298S}$ P510: P < 0.001; Fig 5C and D). Furthermore, FUS$^{ΔNLS/+}$ mice display increased levels of albumin extravasation in comparison with their corresponding WT littermates at P150 (1,025 ± 182 versus 347 ± 103 a.u., respectively; P < 0.0001, Fig 5E and F), including at P270 (1,234 ± 194 a.u. versus 399 ± 148 a.u., respectively; P < 0.0001; Fig 5E and F) with a substantial increase at P450 (1,657 ± 195 a.u. versus 495 ± 75 a.u. in FUS$^{ΔNLS/+}$ versus WT, respectively; P < 0.0001; FUS$^{ΔNLS/+}$ P150 versus FUS$^{ΔNLS/+}$ P450: P < 0.0001; FUS$^{ΔNLS/+}$ P270 versus FUS$^{ΔNLS/+}$ P450: P < 0.01; Fig 5E and F). Similarly, Tbk1$^{+/−}$ mice albumin extravasation was also significantly increased compared with WT littermates at P270 (1,161.0 ± 400 a.u. versus 338.3 ± 86 a.u., respectively; P < 0.0001; Fig 5G and H) and later on at P450 (1,384.0 ± 267 versus 422.0 ± 80 a.u., respectively; P < 0.0001; Fig 5G and H).

Consequently, these data show not only that structural and functional BSCB disruption is a generalized phenomenon discovered in all four ALS mouse models but also that BSCB breakdown is already detectable at stages in which there is no MN loss (P20 in SOD1$^{G93A}$, P150 in FUS$^{ΔNLS/+}$, and P270 and P450 in Tbk1$^{+/−}$ mice).

## Chemogenetic inactivation of MN firing enhances BSCB disruption, whereas stimulation of MN excitation restores BSCB integrity

We therefore established that BSCB disruption is an early event both shared by ALS mouse lines expressing different pathogenic mutations and appreciable before MN loss (in the SOD1$^{G93A}$ and FUS$^{ΔNLS/+}$ mice). We then investigated the pathogenic mechanisms potentially responsible for initiating the BSCB disruption. Loss of excitability in vulnerable MN is among the earliest manifestations of disease (Martinez-Silva et al, 2018), and enhancement of MN excitation has a direct beneficial effect on the burden of misfolded proteins, such as misfolded SOD1, as well as on ER stress and autophagy overload (Saxena et al, 2013; Bączyk et al, 2020). We therefore set out to investigate if early changes in MN excitability may be causally related to the disruption of the BSCB. For this purpose, we exploited an engineered ion channel with orthogonal pharmacology (PSAM/PSEM; [Magnus et al, 2011]), to either stimulate (cation-permeable PSAM: actPSAM) or inactivate (anion-permeable PSAM: inhPSAM) neuronal activity upon administration of the pharmacologically selective synthetic ligand (PSEM$^{308}$ hydrochloride [PSEM$^{308}$]). Intraspinal injections with AAV9 vector (encoding either the actPSAM or the inhPSAM in double-inverted orientation under the human synapsin promoter: hSyn::DIO-actPSAM or hSyn::DIO-inhPSAM.EGFP) were performed in P20 SOD1$^{G93A}$/ChAT-Cre double-tg mice (Fig 7A). Starting from P28, when a robust MN expression of the PSAM was observed, the mice were daily treated for 7 d with PSEM$^{308}$ or with vehicle (Fig 7B). We verified

---

from the two previous images. **(B, D, F, H)** Quantification of the area covered by albumin in the ventral horn of the spinal cord from (B) WT and transgenic SOD1$^{G93A}$ mice (N = 4) at P20, P50, and P80; (D) WT and TDP-43$^{G298S}$ mice (N = 3) at P150, P360, and P510; (F) WT and FUS$^{ΔNLS/+}$ mice (N = 3) at P150, P270, and P450; and WT and Tbk1$^{+/−}$ mice (N = 3) at P270 and P450 (n = 6 sections per mouse). Scale bars: 20 µm. Data information: in (B, D, F, H), data are presented as means ± SD. **P < 0.001, ***P < 0.0001, ****P < 0.0001 (two-way ANOVA [genotype × time point] with Bonferroni correction for multiple comparisons).

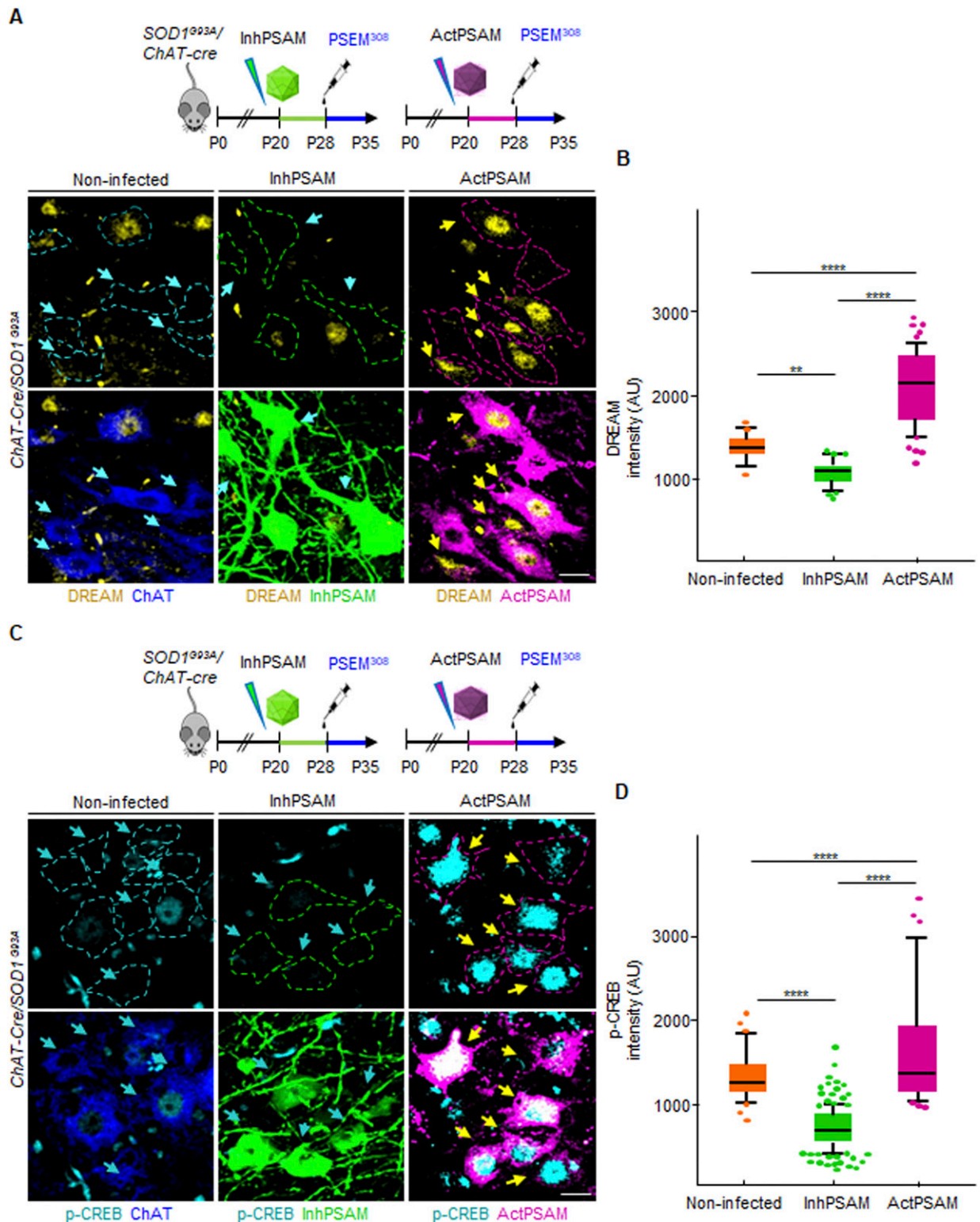

**Figure 6. PSAM/PSEM chemogenetics effectively modify MN activity.**
**(A, C)** Schematic diagram representing the experimental design for control of MN excitation by inhPSAM/PSEM308 (green) and actPSAM/PSEM308 (magenta) on the assessment on (A) DREAM/KChIP3 (yellow) and (C) p-CREB expression in MNs (cyan blue). Dotted lines delimit the border of MNs, which are further detected by ChAT immunostaining (blue). **(A, C)** Cyan blue arrows indicate the absence or less (A) DREAM/KChIP3 and (C) p-CREB staining in noninfected MN nuclei and in inhPSAM/PSEM308-infected MN nuclei (green). **(A, C)** Yellow arrows point to increased levels of (A) DREAM/KChIP3 and (C) p-CREB staining in actPSAM/PSEM308-infected MN nuclei. **(B, D)** Quantification of (B) DREAM/KChIP3 and (D) p-CREB intensity, expressed in a.u. in nuclei of noninfected, inhPSAM/PSEM308 and actPSAM/PSEM308 MNs. The quantifications are represented by box-and-whisker plot; 10–90 percentile is considered. Scale bars: 20 $\mu$m. Data are from N = 5 mice per group. Data information: in (B, D), data are presented as means ± SD. **$P < 0.001$, ****$P < 0.0001$ (one-way ANOVA with Bonferroni correction for multiple comparisons).

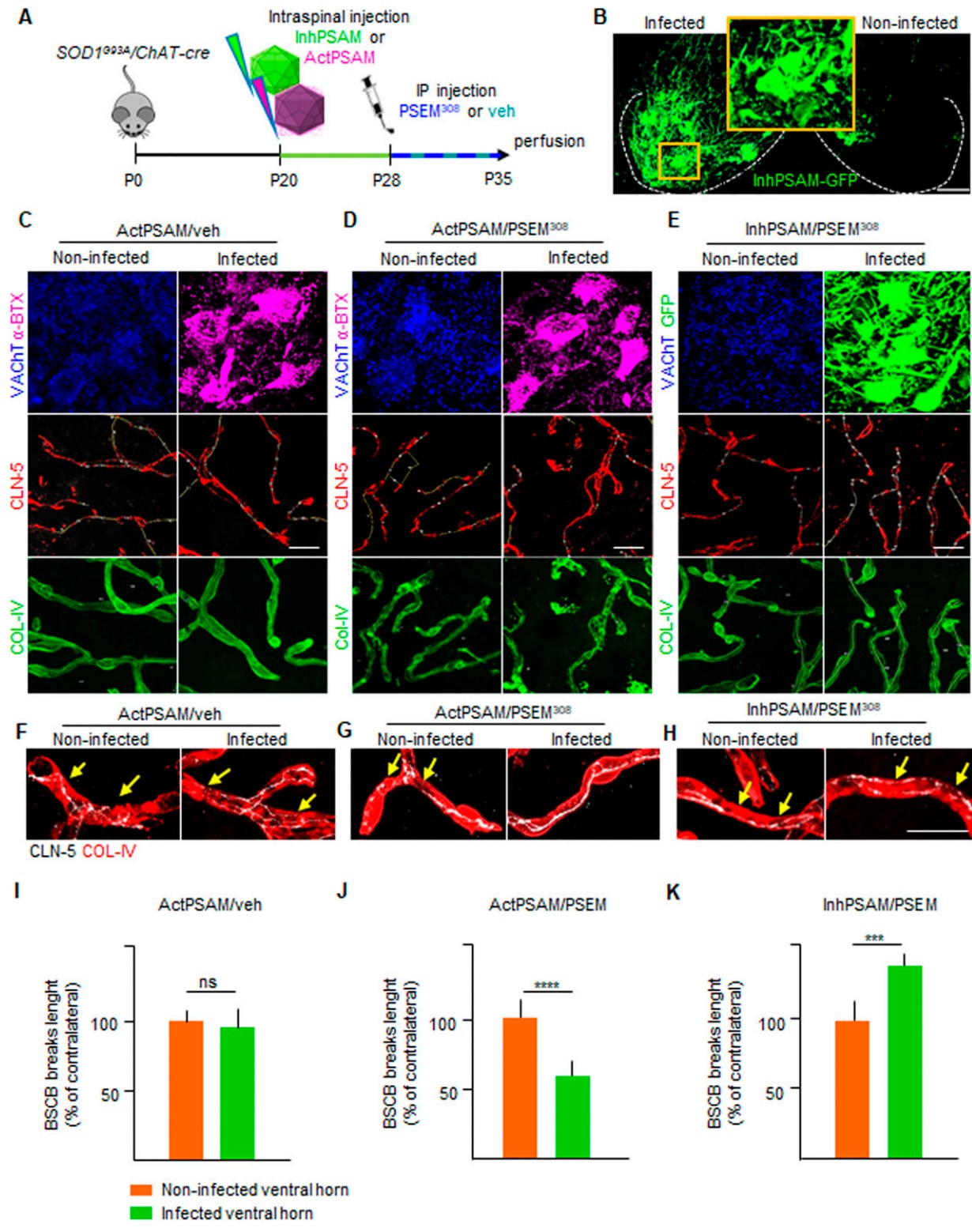

**Figure 7. Inhibition of MN firing increases claudin-5 breaks while MN firing enhancement restores blood–spinal cord barrier impairment.**
**(A)** Experimental design for the chemogenetic control of MN excitation with inhPSAM/PSEM[308] (green) or actPSAM/PSEM[308] (magenta) in *SOD1[G93A]/ChAT-cre* mice injected at P20 and treated with the effector PSEM[308] from P28 until P35. **(B)** Robust MN expression of the inhPSAM (green) upon intraspinal injection of AAV9 in contrast to no expression in the contralateral uninjected side. Dotted lines delineate the boundaries of gray and white matter in the ventral horns of the spinal cord. Scale bar: 50 μm. **(C)** Panel showing chemogenetic expression of actPSAM/veh (treated with vehicle instead of the effector PSEM[308]) in infected MNs (α-bungarotoxin in magenta) in contrast to uninfected MNs stained with VAChT. The panel shows no difference in cumulative breaks length (yellow lines) in CLN-5 ribbon (red) along microvessels

the effectiveness of the chemogenetic modulation on MN firing by showing that actPSAM activation resulted in a significant up-regulation of the activity-dependent gene product DREAM/KChIP3 in the nuclei of infected MNs, compared with the uninfected contralateral MNs (average fluorescence intensity per nucleus: 2,073 ± 475 a.u. versus 1,402 ± 104, respectively; $P < 0.0001$; Fig 6A and B). Likewise, S133-phosphorylated CREB ([pCREB] Wu et al, 2001) was increased in the actPSAM-infected MNs compared with the contralateral uninfected MNs (1,694 ± 723 a.u. versus 1,287 ± 330 a.u., respectively; $P < 0.0001$; Fig 6C and D). Conversely, inhPSAM caused a significant decrease in both activity markers: DREAM (1,043 ± 103 a.u.; inhPSAM versus uninfected; $P < 0.01$; inhPSAM versus actPSAM; $P < 0.0001$, Fig 6A and B) and p-CREB (760 ± 278 a.u.; inhPSAM versus uninfected; $P < 0.0001$; inhPSAM versus actPSAM; $P < 0.0001$; Fig 6C and D) in the infected MNs.

After establishing that chemogenetic approaches have the expected significant impact on the overall activity of infected MNs, we explored the influence of chemogenetic control of MN firing (Fig 7A–K) on the CLN-5 ribbon integrity.

Injection of the AAV9 vector, encoding for the actPSAM, followed by administration of vehicle (instead of the ligand PSEM[308]; Fig 7A and C) did not cause a substantial disruption of the CLN-5 ribbon, compared with the contralateral ventral horn (97.1% ± 20.5% of the uninjected contralateral ventral horn; Fig 7C, F, and I). The experiment proved that intraspinal injection per se did not produce any permanent disruption of the BSCB. Intriguingly, chemogenetic activation of MN firing, by actPSAM/PSEM[308], caused a significant decrease in the BSCB disruption compared with the uninfected ventral horn, with breaks in CLN-5 ribbon reduced up to 59.3% ± 13.1% of contralateral ($P < 0.0001$; Fig 7D, G, and J). By contrast, the inactivation of MN firing by inhPSAM/PSEM[308] resulted in increased fragmentation of the CLN-5 ribbon in the infected ventral horn (129.8% ± 12.7% of contralateral), compared with the noninfected contralateral ventral horn ($P < 0.001$; Fig 7E, H, and K; actPSAM/PSEM[308] versus inhPSAM/PSEM[308]: $P < 0.0001$; Fig 7J and K). These data suggest that BSCB disruption is dependent upon the early excitation-associated dysfunction of MN.

## Chemogenetic activation of astrocytic Gi and Gq signaling is sufficient to restore BSCB integrity

The application of chemogenetic PSAM/PSEM[308] to control MN firing in an ALS mouse model (*SOD1[G93A]*) established a causal link between MN activity and stability of the BSCB in ALS. Two sets of considerations led us to hypothesize that astrocytes may provide a link between neuronal firing and the BSCB in an ALS disease condition: (i) astrocytes sense and respond to increased synaptic

activity and neuronal firing by detecting the release of (among others) ATP, glutamate, and acetylcholine (Durkee & Araque, 2019; Kofuji & Araque, 2020) as well as K[+] ions (Simard & Nedergaard, 2004); (ii) astrocytes are involved in the control of BSCB differentiation and integrity (Abbott et al, 2006; Liebner et al, 2018) by releasing GDNF (Igarashi et al, 1999), angiopoietin-1, and bFGF (Lee et al, 2003). Considering that neuronal activity is monitored by astrocytes via different heteromeric G protein–coupled receptor (GPCR) families and signaling through Gα proteins (Gs, Gi, or Gq; Kofuji & Araque, 2020), we hypothesize that controlling the Gs, Gi, or Gq signaling cascade through designer receptors (DREADDS, coupled to Gs [D(Gs)], Gi [D(Gi)], or Gq [D(Gq)]), we could provoke astrocyte responses related to increased neuronal activity and possibly restore the integrity of the BSCB without intervention on MN themselves. We therefore injected AAV8 encoding GFP alone or citrine-tagged DREADDs coupled to Gs, Gi, or Gq under the control of the GFAP mini promoter (Fig 8A and B) in the spinal cord of P20 *SOD1[G93A]*/*ChAT-Cre* mice (although the expression of Cre is not necessary to target astrocytes, we used the same line as the PSAM/PSEM[308] experiments to maintain a comparable background). Clozapine-N-Oxide (CNO, or vehicle) was administered from P28 until P35 (Fig 8A). The integrity of the CLN-5 ribbon was then evaluated by comparing the injected ventral horn of the spinal cord to the contralateral uninjected ventral horn as an internal control (subject to the same treatments and pharmacology but not receiving the virus). Under these conditions, we verified that DREADDs or GFP was expressed in 65–90% of GFAP[+] cells in the ventral horn of the spinal cord (insert in Fig 8B). When compared with the contralateral ventral horn, the GFP alone in astrocytes (under CNO treatment) did not affect the extent of BSCB breakdown (CLN-5 breaks were 95.8% ± 21.8% of the uninfected contralateral ventral horn; Fig 8C, G, and K). Likewise, expression of DREADDs followed by vehicle treatment did not result in significant changes in BSCB breakdown in injected versus non-injected sides. Taken together, these data confirm that neither CNO treatment nor AAV injection per se affects BSCB integrity. However, when we expressed D(Gi) in astrocytes and administered CNO, the injected side displayed a significant restoration of CLN-5 distribution compared with the contralateral, non-injected side (52.5% ± 12.7% of contralateral; $P < 0.001$; Fig 8D, H, and L). A similar effect was also detected when D(Gq) was activated (49.2% ± 1.3% of the contralateral side; $P < 0.0001$; Fig 8E, I, and M) but not in the case of D(Gs) (103.1% ± 27.8% of the contralateral uninfected side; Fig 8F, J, and N).

We further evaluated the effect of DREADD signaling activation on the astrocytic end-feet coverage of spinal cord capillaries (Fig 9A–G), an additional parameter to evaluate BSCB integrity. Compared with GFP-expressing astrocytes (Fig 9A–C), activation of D(Gi)

---

(identified by COL-IV immunostaining in green) in the uninfected ventral horn, in comparison with the contralateral infected ventral horn (N = 4). **(D)** Panel showing chemogenetic expression of actPSAM/PSEM[308] in infected MNs (α-BTX in magenta) in contrast to uninfected MNs stained with VAChT. Microvessels (green) in proximity of MN activated by actPSAM/PSEM[308] (magenta) display a reduced cumulative breaks length (yellow) of CLN-5 ribbon (red) compared with contralateral uninfected MNs (N = 6). **(E)** Panel displaying chemogenetic expression of inhPSAM/PSEM[308] in infected MNs (immunostained with GFP in green) in contrast to uninfected MNs stained with VAChT. Microvessels in proximity of MNs inactivated by inhPSAM/PSEM[308] (green) display an increase in cumulative breaks length (yellow) of CLN-5 ribbon (red) compared with contralateral uninfected MNs (N = 6). **(F, G, H)** High-magnification view of CLN-5 distribution (white) in the single capillaries (red) of the noninfected and infected ventral horns in inhPSAM/veh, inhPSAM/PSEM[308], or actPSAM/PSEM[308]. Yellow arrows indicate the breaks in the CLN-5 ribbon. Scale bar: 10 μm. **(I, J, K)** Quantification of blood–spinal cord barrier breaks in uninfected and infected ventral horn of the spinal cord of *SOD1[G93A]*/*ChAT-cre* mice subjected to chemogenetic control of activation/inhibition of MN firing. The values are expressed as % of the uninfected contralateral side. Scale bars: 20 μm. Data information: in (I, J, K), data are presented as means ± SD. \*\*\*$P < 0.0001$, \*\*\*\*$P < 0.0001$ (unpaired $t$ test).

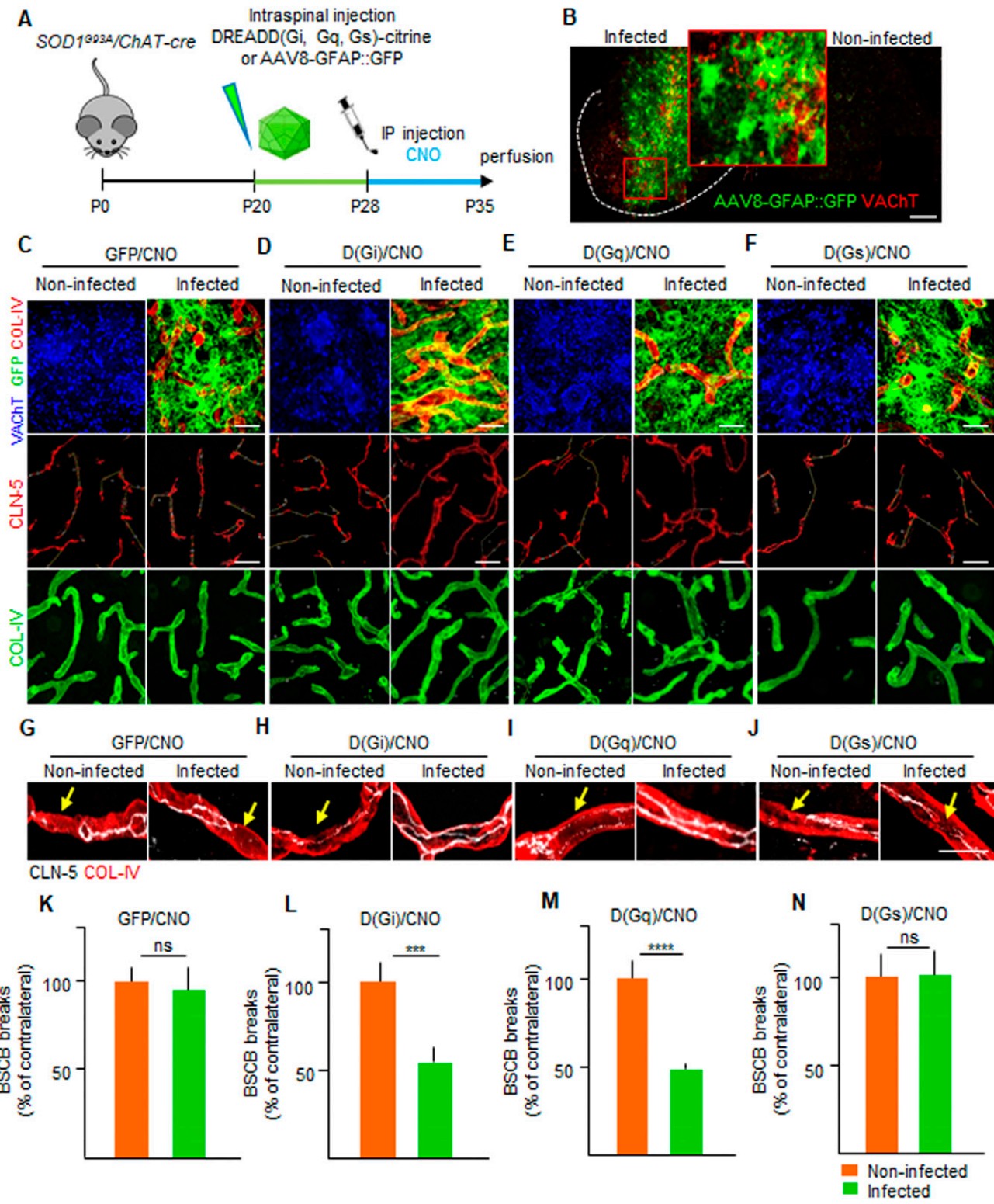

**Figure 8.   Chemogenetic control of astrocytic Gi and Gq signaling restores blood–spinal cord barrier disruption.**
**(A)** Experimental design for the injection of AAV8 encoding GFAP::DREADD-GFP, D(Gi), D(Gq), and D(Gs) in *SOD1^G93A^/ChAT-cre* mice, injected at P20 and treated with the agonist CNO from P28 until P35. **(B)** Pattern of expression of AAV8(GFAP::DREADD-GFP) injected in the ventral horn of the spinal cord and identified by GFP immunostaining (green). The dotted line depicts the boundary of gray and white matter. The insert shows a high magnification of infected astrocytes in the injected ventral horn in contrast to no GFP (astrocytes) staining in the contralateral uninjected ventral horn. Scale bar: 50 μm. **(C)** Panel showing the chemogenetic expression of AAV8(GFAP::GFP) (infected astrocytes in green), followed by CNO administration. The activation does not affect the cumulative breaks length (yellow lines) in CLN-5 ribbon (red) along the

resulted in a significant increase in vascular end-feed coverage (51.2% ± 11.8% in D(Gi) versus 30.7% ± 12.0% in GFP-expressing astrocytes; $P$ < 0.01; Fig 9B and C). Likewise, D(Gq) also increased end-feet coverage (52.9 ± 13.0; $P$ < 0.01 versus GFP; Fig 9B and C), whereas activation of DREADD-(Gs) did not significantly affect the coverage of microvessels by astrocyte end feed (40.5% ± 8.4%; $P$ < 0.1; Fig 9B and C). The effect on astrocyte end feet was additionally confirmed by monitoring the levels of aquaporin 4 (AQP4 [Batavelijc et al, 2014]) surrounding the microvessels, located in the ventral horn of the spinal cord. In animals injected with D(Gi) and treated with CNO, the AQP4+ coverage of microvessels was strongly increased compared with the contralateral uninjected ventral horn of the spinal cord (25.7% ± 6.8% versus 12.6% ± 4.8%, respectively, $P$ < 0.0001; Fig 9D and E). Moreover, we also verified that AQP4+ coverage of vessels also responded to chemogenetic control of MN excitability. Indeed, MN inactivation by inhPSAM, resulted in the significant decrease in AQP4+ coverage of vessels compared with the contralateral ventral horn (7.7% ± 3.1% versus 12.1% ± 4.1%, respectively; $P$ < 0.001; Fig 9F and G).

Taken together, these data show that activation of Gi- and Gq-coupled signaling cascades in astrocytes successfully restores BSCB integrity in presymptomatic *SOD1*[G93A] mice.

### Activation of astrocytic D(Gq) but not D(Gi) ameliorates MN disease markers

Next, we tested if the restoration of BSCB integrity by D(Gi) and D(Gq) correlates with beneficial or detrimental effects on the MN burden of established disease markers (Fig 10A). We considered the accumulation of the misfolded protein SOD1 (using the conformation-specific antibody B8H10; Bosco et al, 2010), ER stress levels (KDEL; Saxena et al, 2013), and the burden of p62+ inclusions (Rudnick et al, 2017; Martinez-Silva et al, 2018; Ouali Alami et al, 2018). The levels of misfolded SOD1 in MN (magenta arrows in Fig 10B), proximal to D(Gq)-activated astrocytes, were significantly lower than those in contralateral MN (orange arrows in Fig 10B [935 ± 346 versus 1,303 ± 342 a.u., respectively, $P$ < 0.0001; Fig 10C]). Likewise, activation of astrocytic D(Gq) decreased the number of p62+ inclusions (1.1% ± 1.5% versus 4.3% ± 4.7% aggregate burden/total cell body, $P$ < 0.0001; Fig 10D and E) and ER stress (1,495 ± 459 versus 1,621 ± 358 a.u., respectively; $P$ < 01; Fig 10F and G) in nearby MNs (magenta arrows), when compared with the contralateral noninfected ventral horn (orange arrows). Surprisingly, activation of D(Gi) in astrocytes did not affect the accumulation of misfolded SOD1 in nearby MN (743 ± 222 versus 730 ± 130 a.u., respectively, $P$ = 0.54; Fig 10H and I) nor p62+ aggregates (4.3% ± 37.2% versus 4.7% ± 36.6% of aggregate burden/

total cell body in the infected versus noninfected ventral horn; $P$ = 0.60; Fig 10J and K) or KDEL levels (1,343 ± 433 versus 1,437 ± 343 a.u., infected versus noninfected ventral horn; $P$ = 0.08; Fig 10L and M). We further extended the investigation of D(Gi) by considering the accumulation of the autophagy marker LC3A. In this case too, D(Gi) did not improve the accumulation of LC3A buildup in MNs (1,475 ± 232 versus 1,431 ± 296 a.u., respectively; Fig 10N and O). Whereas activation of astrocytic D(Gq) drives a generalized amelioration of the disease pathways (MN disease markers and BSCB integrity), stimulation of D(Gi) in astrocytes produces a dissociation between restored BSCB integrity and unmodified disease markers in MN. The D(Gi) findings imply that at this stage, BSCB impairment has limited effect on disease pathways because the BSCB can be restored without impacting ALS pathobiochemistry.

### Restoration of BSCB by astrocytic D(Gi) signaling is independent of MN firing

We then focused on D(Gi), considering that its activation dissociates the effect on BSCB integrality from MN disease markers, which could potentially shed light on the role of BSCB disruption on the pathogenesis of MN disease. We therefore investigated the mechanism involved in astrocytic D(Gi)-mediated restoration of BSCB. Because increasing MN firing was sufficient to improve BSCB integrity, we explored whether astrocytic Gi would prove beneficial by modulating MN activity, for example, by inducing the release of excitatory neurotransmitters (Durkee et al, 2019). In this case, we reasoned that any beneficial effect of D(Gi) on BSCB would be significantly decreased by the concomitant inactivation of MN firing. We arranged a multiplexed chemogenetic strategy to inactivate MN firing using the inhPSAM/PSEM[308], while simultaneously triggering Gi signaling in astrocytes by D(Gi) (Fig 11A and B). To this end, intraspinal injections were performed in *SOD1*[G93A]/*ChAT-Cre* mice with AAV9 encoding for hSyn::DIO-inhPSAM (expression restricted to MN) and with AAV8 encoding GFAP::D(Gi) (expression restricted to astrocytes). After 10 d, we administered to the animals both CNO and PSEM[308] (or vehicle + PSEM[308]) for a further 7 d (Fig 11B). Inactivation of MN alone resulted in the anticipated increased disruption of the BSCB (129.8% ± 8.7% of the contralateral noninfected ventral horn; $P$ < 0.01; Fig 11C, E, and G). However, the activation of D(Gi) in the presence of MN inactivation still resulted in a substantial improvement of BSCB integrity (68.6% ± 10.7% of the contralateral uninfected ventral horn; $P$ < 0.01; Fig 11C, E, and G). Nevertheless, MN firing inactivation by inhPSAM/PSEM[308], even when co-applied with astrocytic D(Gi) activation, caused increased levels of LC3A (1,840 ± 396 versus 1,693 ± 395 a.u., respectively, in the

---

COL-IV+ vessels (green) in the infected ventral horn of the spinal cord compared with the contralateral uninfected ventral horn. **(D)** Panel displaying the activation of D(Gi)/CNO in astrocytes. The activation of astrocytic Gi signaling causes the reduction of CLN-5 break (yellow lines) burden in the infected horn compared with the contralateral uninfected horn. **(E)** Panel displaying representative pictures of D(Gq)/CNO activation in astrocytes, resulting in decreased CLN-5 break (yellow lines) burden in the infected ventral horn. **(F)** Representative pictures showing activation of D(Gs)/CNO in infected astrocytes. D(Gs) activation does not modify CLN-5 break (yellow lines) burden when compared with the contralateral uninfected ventral horn. **(C, D, E, F)** MNs in the uninjected ventral horn are detected by VAChT immunostaining (blue). Infected astrocytes, identified by GFP (green), are co-stained with COL-IV (red) to visualize the vasculature surrounding the astrocytes in the injected ventral horn. **(G, H, I, J)** High-magnification view of CLN-5 distribution (white) in the single microvessels (red), displaying the effect of control GFP/CNO, D(Gi)/CNO, D(Gq)/CNO, and D(Gs)/CNO activation in infected and noninfected ventral horns. Yellow arrows indicate the breaks in the CLN-5 ribbon. Scale bar: 10 μm. **(K, L, M, N)** Quantification of blood–spinal cord barrier breaks in uninfected and infected ventral horns of the spinal cord of *SOD1*[G93A]/*ChAT-cre* mice. Data are from N = 8 mice per group of experiments subjected to injection of AAV8(GFAP::GFP)/CNO, D(Gi)/CNO, D(Gq)/CNO, and D(Gs)/CNO. The values are expressed as % of the uninfected contralateral side. Scale bars: 20 μm. Data information: in (K, L, M, N), data are presented as means ± SD. ***$P$ < 0.0001, ****$P$ < 0.0001 (unpaired $t$ test).

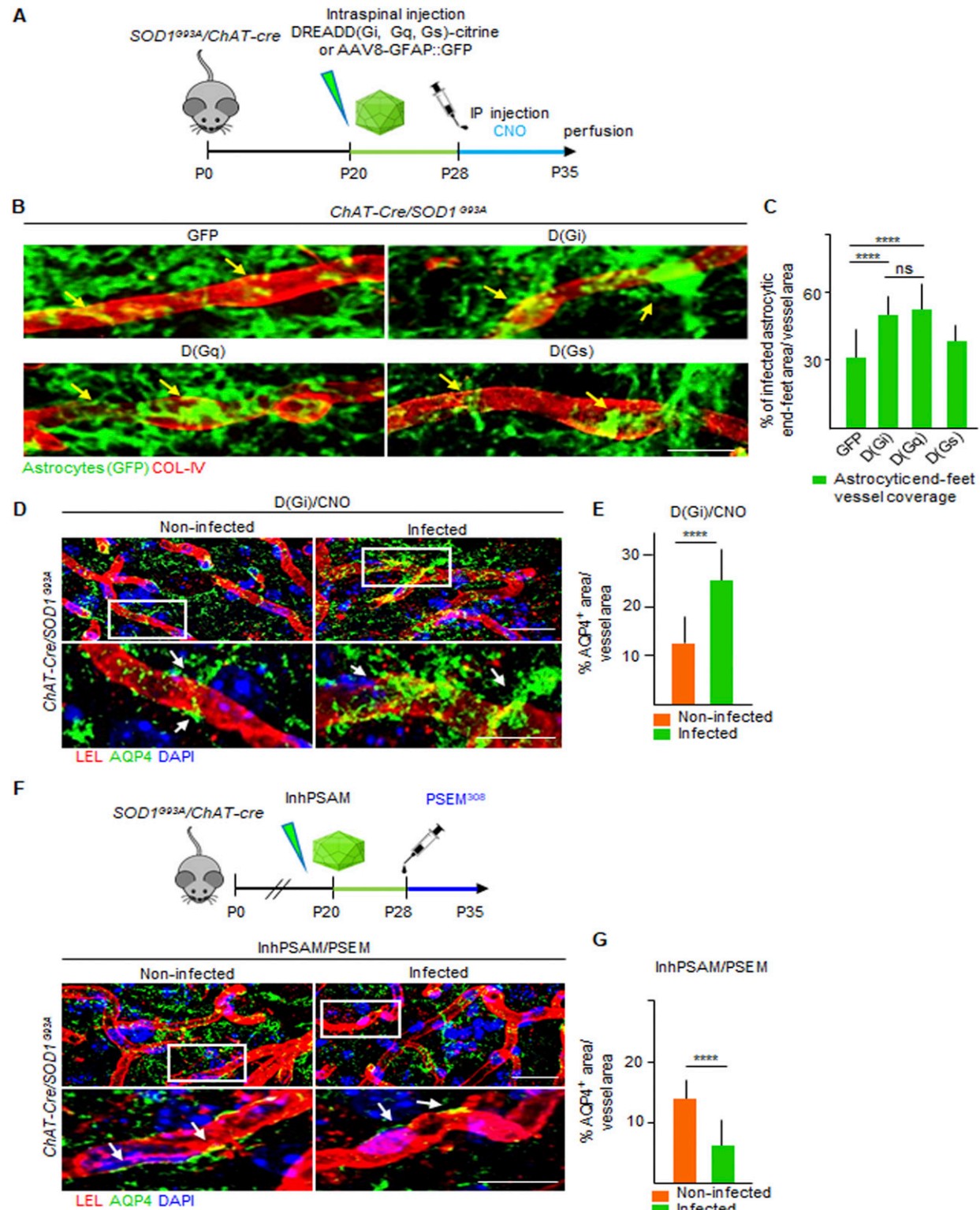

**Figure 9. Astrocytic end-feet coverage of microvessels is enhanced by D(Gi) and D(Gq) activation, while diminished after MN firing inhibition.**
**(A)** Experimental design for the injection of AAV8 encoding GFAP::DREADD-GFP, D(Gi), D(Gq), and D(Gs) in *SOD1$^{G93A}$/ChAT-cre* mice, injected at P20 and treated with the agonist CNO from P28 until P35. **(B)** High magnification of single microvessels (red) showing astrocytic end-feet coverage (green) after GFP, D(Gi), D(Gq), and D(Gs) activation. The pattern of infected astrocytes (identified by GFP immunostaining in green) on the vessels significantly increases in D(Gi)/CNO, D(Gq)/CNO, and (DGs)/CNO compared with GFP/CNO alone. Scale bar: 7 μm. **(C)** Quantification of end-feet coverage of vessels after DREADD treatments, expressed as % of the GFP⁺ area on the total vessel area. **(D)**. Representative pictures of aquaporin-4 (green) expression localized in the astrocytic end-feet enveloping microvessels (red) in the ventral horn of the

co-injected ventral horns versus contralateral uninjected horns; $P <$ 0.0001; Fig 11H) and a higher burden of p62[+] inclusions (4.6% ± 3.1% versus 3.2% ± .1.8% of aggregate burden/total cell body, coinfected versus contralateral; $P < 0.1$; Fig 11I), confirming the effective target engagement of inhPSAM/PSEM[308] (Saxena et al, 2013) and further demonstrating that BSCB integrity, and disease marker burden in MN can be dissociated. We further explored whether D(Gi) activation would confer an additional beneficial effect when applied together with increased MN firing (i.e., if the two beneficial effects are additive or not). We compared the CLN-5 ribbon integrity in spinal cord sections in which both MN firing (actPSAM/PSEM[308]) and astrocytic D(Gi) were activated. Notably, single chemogenetic and multiplexed chemogenetic samples displayed the same degree of BSCB restoration, (actPSAM/PSEM[308] + D(Gi)/CNO: 60.9% ± 6.7% of contralateral noninfected versus 59.2% ± 13.1% in actPSAM/PSEM[308] + D(Gi)/veh; $P > 0.05$; Fig 11D, F, and J). Driving MN excitation by actPSAM/PSEM[308] therefore not only mimicked but also truly occluded the effect of astrocytic Gi signaling.

Taken together, the double-chemogenetic experiments imply that the consequences of D(Gi) on BSCB are independent of MN firing, although they mimic the effects of MN activity stimulation, suggesting that astrocytic Gi signaling may be located downstream of MN excitation.

### Astrocytic Wnt7a and Wnt5a levels are enhanced by Gi signaling in astrocytes and suppressed by MN inactivation, in correlation with BSCB modifications

We sought to identify the mediator(s) involved in the sealing of the BSCB induced by the activation of Gi signaling in astrocytes. We focused on the Wnt family because these mediators are necessary in the establishment of the blood-brain-barrier (BBB) during development (Liebner et al, 2008), as well as in its maintenance in adulthood (Tran et al, 2016; LeBlanc et al, 2019). First, we explored if the blockade of Wnt signaling modified the integrity of the BSCB in WT and *SOD1[G93A]* mice. The animals were treated for 14 d either with vehicle or with the porcupine (PORCN) inhibitor (which blocks Wnt acylation and release) C59 (Proffitt et al, 2013; Torres et al, 2019), starting from day P20. The administration of the Wnt antagonist C59 caused a small disruption of the BSCB in the WT mice (124.7% ± 16.4% of WT control mice treated with vehicle) and a substantial worsening of the BSCB integrity in the *SOD1[G93A]* mice (157.2% ± 17.7% of WT veh; $P < 0.0001$; Fig 12A and B), indicating that Wnt proteins are involved in balancing the BSCB integrity in the ALS mice.

Next, we focused on Wnt5a, the most abundant Wnt family member in the spinal cord (Ouali Alami et al, 2018), recognised as sufficient for inducing a BBB-like phenotype in endothelial cells (Artus et al, 2014) and on Wnt7a, a critical mediator in the

establishment and maintenance of the BBB phenotype (Cho et al, 2017; Wang et al, 2018). We exploited single-molecule mRNA in situ hybridization (coupled to immunostaining for the identification of cellular subpopulations) to quantify Wnt5a and Wnt7a in spinal cord astrocytes (identified by GFAP staining). Both Wnt5a and Wnt7a mRNA molecules were readily detected in GFAP[+] astrocytes in the WT and *SOD1[G93A]* spinal cord sections (Fig 12C–F). Notably, both astrocytic Wnt5a mRNA molecules (58.6% ± 12.3% of WT; $P < 0.01$; Fig 12C and D) and Wnt7a mRNA molecules (28.6% ± 11.9% of WT; $P < 0.0001$; Fig 12E and F) were significantly decreased in the *SOD1[G93A]* mice compared with WT controls.

We then explored if Wnt5a and Wnt7a levels could be modulated by chemogenetic manipulations in correlation with the integrity of the BSCB. We determined that chemogenetic inactivation of MN firing by inhPSAM/PSEM[308] caused a significant additional decrease in Wnt5a mRNA (40.5% ± 13.1% of contralateral; $P < 0.0001$; Fig 12G and H) and in the Wnt7a mRNA molecules (71.9% ± 11.6% of the uninfected contralateral horn; $P < 0.01$; Fig 12I and J) in the astrocytes of the infected ventral horn compared with the contralateral side. These data suggest that disruption of the BSCB driven by MN inactivation is associated with the reduced expression of Wnt5a and Wnt7a in astrocytes. We tested this hypothesis by assessing the impact of astrocytic Wnt5a and Wnt7a levels on D(Gi) activation in astrocytes. We obtained a strong increase in Wnt5a (180.8% ± 33.3% of contralateral; $P < 0.01$; Fig 12K and L) and Wnt7a mRNA molecules (219.4% ± 6.4%; $P < 0.01$; Fig 12M and N) in the infected astrocytes compared with the uninfected astrocytes in the contralateral side. Wnt5a and Wnt7a levels were therefore highly correlated with the integrity of BSCB during the pharmacological inhibition of Wnt signaling, as well as with an increase in MN firing and stimulation of astrocytic Gi signaling.

### Activation of astrocytic Gi signaling at later stages ameliorates both BSCB disruption and disease burden

Because ALS pathogenic processes are heterogeneous and evolve over time (Ouali Alami et al, 2018), we investigated whether prolonged D(Gi) activation would enable the restoration of the BSCB in correspondence to the first wave of MN denervation (taking place at about P50; [Pun et al, 2006]). *SOD1[G93A]* mice were injected with AAV8(GFAP::D(Gi)-Citrine) at P20 and treated with a single injection CNO daily for 20 d, from P30 until P50 (Fig 13A). We observed that prolonged activation of astrocytic D(Gi) resulted in a significant decrease in the breakdown of the BSCB (CLN-5 breaks length, 59.9% ± 8.0% in D(Gi) versus uninfected contralateral controls; $P < 0.001$; Fig 13B and D). Surprisingly, the prolonged activation of astrocytic D(Gi) affected the burden of MN disease markers. The MN nearby D(Gi)-expressing astrocytes displayed a reduced burden of p62[+]

---

spinal cord of SOD1[G93A]/ChAT-cre. Nuclei are immunostained with DAPI (blue). Inserts show high magnification of AQP4 (green) labeling, colocalized with a single microvessel identified by COL-IV (red) staining. Scale bars: 7 $\mu$m. **(E)** Quantification of the AQP4[+] area surrounding the vessels in the *SOD1[G93A]/ChAT-cre* mice injected with D(Gi)/CNO (N = 3). Values are expressed as % of the total vessel area. **(F)** Experimental design displaying MN firing inactivation via inhPSAM/PSEM[308] in SOD1[G93A]/ChAT-cre mice. The inhibition of MN firing reduces AQP4 expression at the level of astrocytic end-feet coverage surrounding the vessels, localized in the infected ventral horn compared with the uninfected contralateral horn. Inserts show high magnification of AQP4 immunostaining (green) colocalized with single microvessels (red). White arrows indicate the distribution of AQP4. Scale bar: 7 $\mu$m. **(G)** Quantification of the AQP4[+] area surrounding the vessels in the *SOD1[G93A]/ChAT-cre* mice injected with inhPSAM/PSEM[308]. Data are from N = 3 mice per group of experiments. Values are expressed as % of the total vessel area. Scale bars: 20 $\mu$m. Data information: in (C, E, G), data are presented as means ± SD. **(C, E, G)** ****$P < 0.0001$ (one-way ANOVA with Bonferroni correction for multiple comparisons [C] and unpaired $t$ test [E, G]).

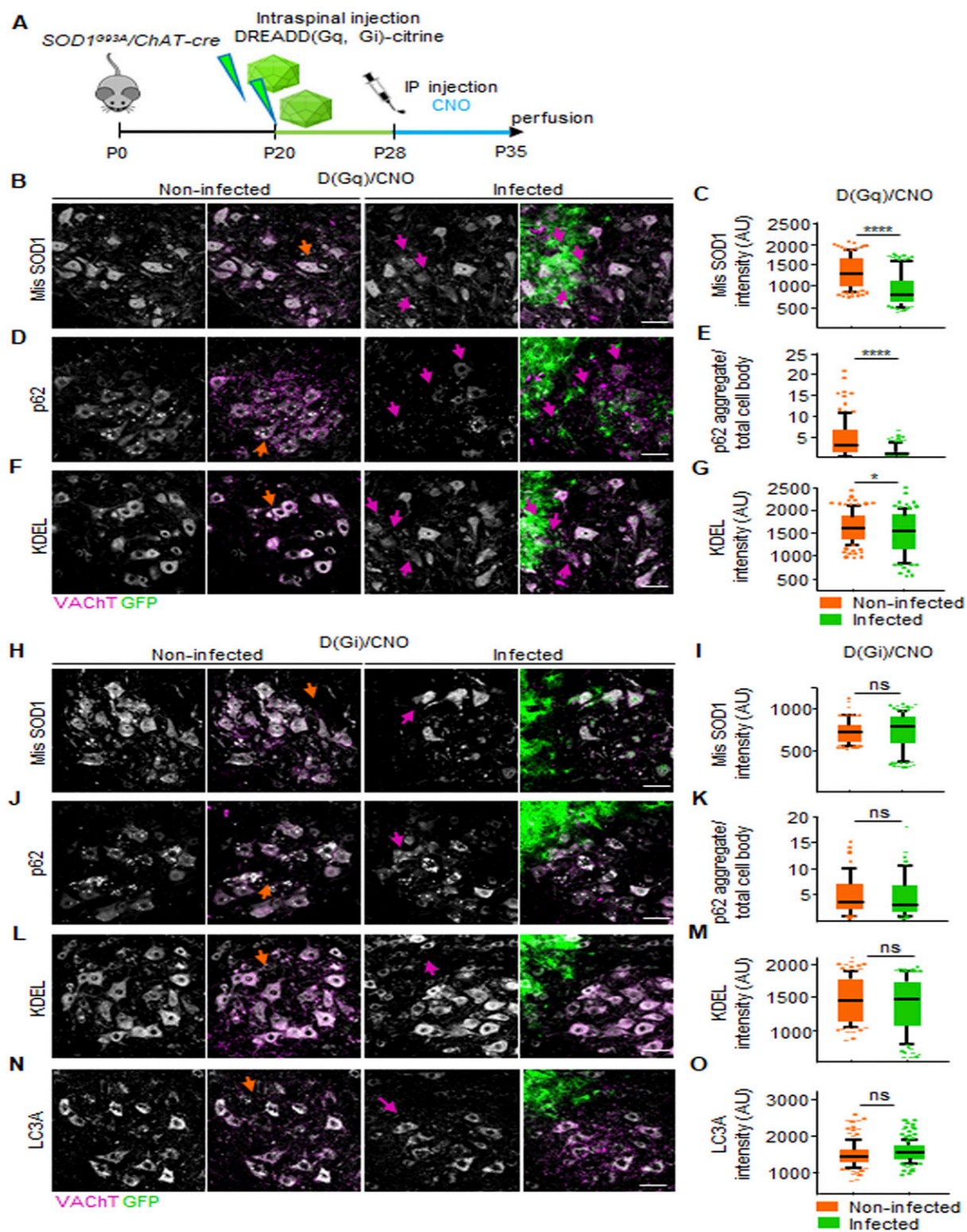

**Figure 10. Chemogenetic activation of astrocytic Gq, but not Gi, signaling decreases the burden of disease markers in MN.**
**(A)** Experimental design for the injection of AAV8 encoding D(Gq) and D(Gi) in *SOD1^G93A^/ChAT-cre* mice, at P20 and treated with the agonist CNO from P28 until P35. **(B, D, F)** Representative pictures of (B) misfolded SOD1 (white), (D) p62 (white), and (F) KDEL (white) immunostaining in MN located close to the activated astrocytes (green) in the infected ventral horn and in MN in the contralateral uninfected ventral horn of *SOD1^G93A^/ChAT-cre* spinal cords subjected to AAV8 D(Gq)/CNO injection. MNs are identified by VAChT immunostaining (magenta), and infected astrocytes are detected by GFP (green). **(B, D, F)** Orange arrows indicate the overload of (B) misfolded SOD1 (white), (D) p62 (white), and (F) KDEL (white) burden in the MNs located in the noninfected ventral horn, and magenta arrows point to less accumulation of (B) misfolded

aggregates (11.1% ± 10.1% versus 20.4% ± 54.9% in uninfected contralateral MN, aggregate burden/total cell body; $P < 0.01$; Fig 13E and F) and decreased levels of LC3A (1,657 ± 386 a.u. versus 2,180 ± 485 a.u., respectively; $P < 0.0001$; Fig 13E and G).

At the same time, the amount of BSCB restoration remained substantial even upon simultaneous MN inactivation by inhPSAM (CLN-5 breaks length, 72.7% ± 8% of uninfected contralateral side; $P < 0.0001$; Fig 13H–L), confirming the independence of the astrocytic D(Gi) from MN excitability effects.

In conclusion, although BSCB restoration, driven by Gi signaling in astrocytes at early stages of the disease, was not associated with any effect on disease markers, it could be linked to neuroprotective effects at later phases of ALS pathology.

# Discussion

In the present work, we have showed that (i) BSCB structural and functional disruption is detected in four ALS mouse lines with unrelated mutations and different degrees of MN loss; (ii) the disruption of the BSCB takes place before any MN loss (in at least two lines), and it is restored by enhancing MN excitability; and (iii) the activation of Gi signaling in astrocytes reverts the disruption of the BSCB by increasing Wnt5a/Wnt7a expression but without impacting the burden of disease markers in MN and independently of MN firing. Overall, these data suggest first that in healthy conditions, MN excitability controls BSCB integrity through astrocytes and second that in ALS, the loss of MN excitability and firing leads (possibly through the decrease in signaling driven by the Gi-coupled receptor) to the down-regulation of Wnt5a/7a in astrocytes, which in turn provokes the opening of the BSCB. Whereas at the very initial stages, the restoration of BSCB integrity does not affect MN pathobiochemistry, the opposite is true later on: the disruption of the BSCB is an event secondary to MN dysfunction, which then feeds back on MN and aggravates the ongoing pathogenic process.

Although the disruption of the BSCB in ALS has been previously reported (Garbuzova-Davis et al, 2007a, 2007b; Zhong et al, 2008), the nature of the primum movens has remained debated: Is BSCB impairment driven by MN dysfunction or is it a MN-independent event? Some (controversial) evidence has pointed toward an MN-independent origin: Endothelial cells expressing mutant *SOD1* in vitro display a cell-autonomous disruption of TJs because of the misfolded protein itself (Meister et al, 2015). At the same time, endothelial cell–selective excision of the mutant *SOD1* transgene

does not prevent the breakdown of the BSCB nor affect the survival of the transgenic mice (Zhong et al, 2009). The loss of pericytes observed in spinal cord samples from ALS patients has been hypothesized to be at the origin of BSCB disruption (Winkler et al, 2013). However, increased PDGF-C signaling appears to be associated with BSCB disruption in ALS (Liebner et al, 2008), even if PDGFR-$\alpha$ signaling is actually necessary for the survival of these cells (Kisler et al, 2017). Recently, microglial activation, alone or together with peripheral immune cells contribution, has been hypothesized to be involved in the opening of the BSCB (Puentes et al, 2016; Frakes et al, 2017; Epperly et al, 2019). We have demonstrated that chemogenetic control of MN firing can bidirectionally modulate BSCB integrity: Increased firing reduces BSCB disruption, whereas neuronal inactivation worsens it. This pattern corresponds to what has been observed for other disease markers such as misfolded SOD1 accumulation, ER stress, and unfolded protein response (Saxena et al, 2013; Bączyk et al, 2020). Even if other cell types play contributing roles, the impairment of the BSCB is thus the manifestation of a pathogenic process, primarily originating in MN. Notably, BSCB impairment is detected in ALS mouse models before any MN loss (such as in the $FUS^{\Delta NLS}$ and $SOD1^{G93A}$) but at stages when hypoexcitability of vulnerable MN is already detectable (Martinez-Silva et al, 2018). Thereafter, the disruption of the BSCB may be a consequence of early MN dysfunction.

How is this dysfunction translated into a vascular phenotype? Considering that astrocytes are sensitive to neuronal activity (through several glutamatergic, GABAergic, cholinergic, and purinergic GPCR; Kofuji & Araque, 2020) and, at the same time, they are key organizers of the BSCB (Abbott et al, 2006), these cells become prime candidates for investigation. We exploited DREADDs with astrocyte-selective promoters to obtain a direct cell- and time-specific modulation of GPCR signaling (Orr et al, 2015; Bang et al, 2016; Adamsky et al, 2018; Jones et al, 2018; Nagai et al, 2019). We demonstrated that D(Gq) and D(Gi) can induce the restoration of BSCB integrity, although D(Gq) itself appears to be beneficial toward all disease readouts, suggesting that it may restore the BSCB by ameliorating MN firing, possibly upon triggering gliotransmitter release (Durkee et al, 2019). Critically, D(Gi) signaling acts directly on the BSCB, independently of MN firing, as we have shown in our multiplexed chemogenetic strategy.

How does astrocytic D(Gi) induce the restoration of the BSCB? We elected to investigate the role of Wnt proteins, in particular Wnt5a and Wnt7a. Wnt family members are involved in the establishment of the BBB during development (Liebner et al, 2008; Stenman et al, 2008) and contribute to its maintenance in the adult brain (Artus

---

SOD1 (white), (D) p62 (white), and (F) KDEL (white) burden in MN close to Gq activated astrocytes (green). **(C, E, G)** Quantification of (C) the intensity of misfolded SOD1, (E) p62 aggregates per total cell body, and (G) KDEL intensity in MNs located in the infected and uninfected contralateral ventral horns of mice subjected to chemogenetic activation of astrocytes via D(Gq)/CNO. **(H, J, L, N)** Representative pictures of (H) misfolded SOD1 (white), (J) p62 (white), (L) KDEL (white), and (N) LC3A (white) immunostaining in MN located close to infected astrocytes (green) in the infected ventral horn and in MNs in the contralateral uninfected ventral horn of $SOD1^{G93A}$/*ChAT-cre* spinal cords subjected to AAV8 D(Gi)/CNO injection. **(H, J, L, N)** Orange arrows indicate the overload of (H) misfolded SOD1 (white), (J) p62 (white), (L) KDEL (white), and (N) LC3A (white) burden in the MNs located in the noninfected ventral horn, while magenta arrows point to decreased levels of (H) misfolded SOD1 (white), (J) p62 (white), (L) KDEL (white), and (N) LC3A (white) burden in MN close to Gi-activated astrocytes (green). The first column of each experiment shows MN markers alone, and the second column displays the co-immunostaining with VAChT or the co-immunostaining with VAChT and GFP (for infected astrocytes). **(I, K, M, O)** Quantification of (I) the intensity of misfolded SOD1, (K) p62 aggregates per total cell body, (M) KDEL, and (O) LC3A intensity in MNs located in the infected and uninfected contralateral ventral horns of mice subjected to chemogenetic activation of astrocytes via D(Gi)/CNO. The quantifications are represented by a box-and-whisker plot; 10–90 percentile is considered. Data are from N = 6 mice per group of experiments. Scale bars: 20 $\mu$m. Data information: in (C, E, G, I, K, M, O), data are presented as means ± SD. *$P < 0.1$, ****$P < 0.0001$ (unpaired $t$ test).

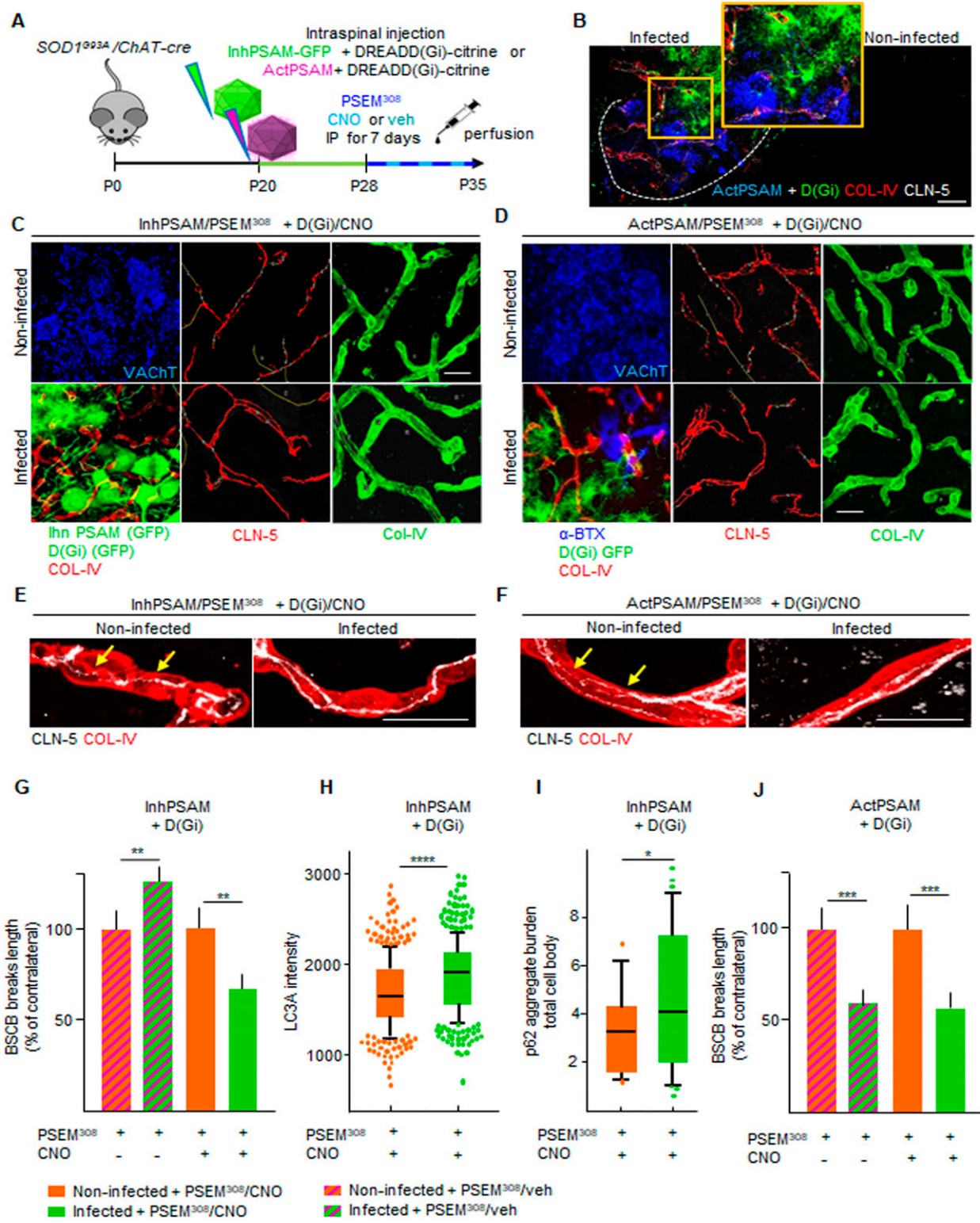

**Figure 11. Blood–spinal cord barrier (BSCB) restoration by astrocytic D(Gi) signaling is independent of MN excitation.**
**(A)** Experimental design for the intraspinal injection of AAV9(hSyn::DIO-inhPSAM or -actPSAM) in combination with AAV8(GFAP::D(Gi)-Citrine) for the multiplexed chemogenetic experiments in the $SOD1^{G93A}/ChAT$-$cre$ mice at P20 and treated with the effector PSEM[308] and the agonist CNO or vehicle from P28 until P35. **(B)** Representative picture of intraspinal injection of AAV9 encoding actPSAM, highly expressed in MNs (blue), in combination with injection of AAV8 D(Gi), highly specific for astrocytes (green) in the infected ventral horn in contrast to no expression pattern in the uninjected ventral horn. The dotted line depicts the boundary of gray and white matter. The insert shows a high magnification of infected MNs and astrocytes in the injected ventral horn. Scale bar: 50 $\mu m$. **(C)** Representative panel displaying

et al, 2014; Tran et al, 2016). Low Wnt/β-catenin signaling actually characterizes the site in the brain with the lowest integrity of the BBB and the highest permeability (Benz et al, 2019). Among Wnt family members, Wnt7a appears to be critical during development (Daneman et al, 2009) and maintains the integrity of the BBB through Gpr124, during a stroke or in glioblastoma (Chang et al, 2017), as well as in homeostatic conditions. Nevertheless, Wnt5a is sufficient to induce a BBB phenotype in endothelial cells in vitro (Artus et al, 2014), and it is among the most strongly expressed Wnt members in astrocytes (Ouali Alami et al, 2018). We have demonstrated that Wnt5 expression in astrocytes is reduced early on in *SOD1^G93A* mice, and the suppression of Wnt signaling by the porcupine inhibitor C59 further worsens BSCB disruption. Moreover, the astrocytic Wnt5a mRNA and Wnt7a mRNA levels bidirectionally correlate with the integrity of the BSCB, being down-regulated upon inactivation of MN firing (worsening of the BSCB integrity) and up-regulated by D(Gi) activation in astrocytes (ameliorating the BSCB integrity). During increased availability of Wnt5a and Wnt7a, it is believed that the activation of the β-catenin signaling cascade in endothelial cells is a key event in inducing and maintaining BSCB specializations. Canonical Wnt signaling through β-catenin is sufficient to induce a BBB-like phenotype in cultured endothelial cells (Laksitorini et al, 2019), and it is critically involved in establishing the BBB phenotype in vivo (Liebner et al, 2008; Chang et al, 2017; Cho et al, 2017; Wang et al, 2018), whereas reduced Wnt/β-catenin signaling spatially corresponds to sites of high BBB permeability in the brain (Benz et al, 2019). Moreover, noncanonical Wnt signaling may contribute to the establishment of the BBB phenotype (Pinzón-Daza et al, 2014). Sox17 has been identified as a major target of the β-catenin pathway in endothelial cells (Corada et al, 2019), together with Lef1 and Ets1 (Roudnicky et al, 2020). These Wnt-regulated transcription factors up-regulate the transcription of several junctional proteins, including VE-cadherin and claudin-5 (Roudnicky et al, 2020), possibly involved in the repair of the BSCB. Furthermore, claudin-5 is dynamically internalized and stored in intracellular vesicles upon inflammatory stimuli and trafficked back to the cell surface when the inflammatory signaling is subsided (Stamatovic et al, 2003, 2006, 2009). It is therefore conceivable that transcriptional mechanisms and trafficking may be responsible for Wnt5a/7a-driven BSCB restoration.

In principle, it is possible that additional mediators may contribute to the D(Gi) effect (e.g., GDNF; Igarashi et al, 1999). One could also hypothesize that the up-regulation of Wnt5a/7a induced by D(Gi) may affect other nonneuronal cells (e.g., microglia) and only indirectly play a role in the integrity of the BSCB. Although this model is theoretically possible and worth further investigation, the direct effects of Wnt5a/7a on BSCB integrity are well established (Liebner et al, 2008; Stenman et al, 2008; Artus et al, 2014; Tran et al, 2016; Cho et al, 2017; Wang et al, 2018; Benz et al, 2019; Laksitorini et al, 2019; LeBlanc et al, 2019). Because microglia do respond to astrocytic Wnt (Ouali Alami et al, 2018), Wnt signaling is thought to cause microglial proliferation and substantial pro-inflammatory activation (Halleskog et al, 2012, 2011), although the ultimate outcome may depend on the context (Halleskog and Schulte, 2013). Putative indirect actions of astrocytic D(Gi) remain to be further investigated.

In our current interpretative model (Fig S5), MN activity is hypothesized to be linked to BSCB integrity via astrocytes and astrocyte-secreted Wnt5a/7a. Nevertheless, to date, virtually nothing is known about the MN-originated signals that modulate Wnt5a/7a in adult astrocytes. However, one can speculate about the role of astrocytic GPCRs involved in monitoring neuronal and synaptic activities (e.g., GABA, glutamate, ATP, and acetylcholine receptors are expressed by astrocytes; Kofuji & Araque, 2020). Insufficient engagement of these receptors due to MN hypoexcitability and reduced synaptic excitation (Martinez-Silva et al, 2018; Bączyk et al, 2020) may also decrease Wnt5a/7a in astrocytes. One could further speculate that MN-initiated signaling might converge on NF-kB because overactivation of IKK-β in astrocytes is sufficient to increase Wnt5a expression (Ouali Alami et al, 2018).

The ability to control BSCB integrity in vivo enables us to address a critical issue in the vascular biology of ALS: Is the disruption of the BSCB detrimental to the MN disease process? Previous evidence suggests that closure of the BSCB by APC administration is associated with better outcomes; however, this effect was linked in reality to the decrease in the expression of the mutant *SOD1* gene itself induced by APC (Zhong et al, 2009). Likewise, the worsening of disease progression observed upon warfarin administration has been attributed to the substantial disruption of the BSCB, the extravasation of erythrocytes, and iron-triggered radical oxygen toxicity (Winkler et al, 2014). Nevertheless, this interpretation is complicated by the unclear target selectivity of warfarin itself and by the lack of an identified cellular target. There is further proof that even the substantial disruption of the BSCB that follows NF-κB activation in astrocytes is not necessarily associated with a worsening of the disease progression (Ouali Alami et al, 2018). Furthermore, mutant *SOD1* mice lacking aquaporin-4 display an intact BSCB

immunostaining for the multiplexed chemogenetic experiments in which MN firing inhibition via inhPSAM/PSEM[308] (infected MNs in green) combined with astrocytic Gi activation diminished the breaks (yellow lines) in CLN-5 ribbon (red) in the infected ventral horn compared with the uninfected contralateral ventral horn. **(D)** Panel showing immunostaining for the multiplexed chemogenetic experiment in which MN firing enhancement via actPSAM/PSEM[308] (infected MNs detected by α-BXT in blue) combined with astrocytic Gi activation (astrocytes identified by GFP in green) further decreases breaks (yellow lines) along CLN-5 ribbon (red). MNs in the noninfected ventral horn are identified by VAChT immunostaining, whereas vessels are labeled with COL-IV (red in the combined picture and green in the single picture). **(D, E, F)** High-magnification view of CLN-5 distribution (white) along the single capillaries (red) in the multiplexed chemogenetics (E) inhPSAM/PSEM[308] + D(Gi)/CNO and in (D) actPSAM/PSEM[308] + D(Gi)/CNO. **(D, E)** Yellow arrows indicate the discontinuity of CLN-5 ribbon (white) in the noninfected ventral horn in contrast to a more homogeneous distribution in the infected ventral horn of (E, D) both experiments. Scale bar: 10 μm. **(G)** Quantification of BSCB disruption expressed as % of breaks of the contralateral uninfected ventral horn after multiplexed chemogenetic experiments with inhPSAM/PSEM[308] + D(Gi)/CNO (CNO +) or with inhPSAM/PSEM[308] + D(Gi)/veh (CNO − [column with milled pattern]). **(H, I)** Quantification of MN (H) LC3A intensity and (I) p62 aggregates per cell body in the double-infected (inhPSAM/PSEM[308] + D(Gi)/CNO) ventral horn compared with the uninfected ventral horn. **(J)** Quantification of BSCB disruption expressed as % of breaks of the contralateral uninfected ventral horn after multiplexed chemogenetic experiments with actPSAM/PSEM[308] + D(Gi)/CNO (CNO +) or with actPSAM/PSEM[308] + D(Gi)/veh (CNO − [column with milled pattern]). **(H, I)** The quantifications of MN markers in (H, I) are represented by the box-and-whisker plot; 10–90 percentile is considered; N = 4–6 mice per group of experiments. Scale bars: 20 μm. Data information: in (G, H, I, J), data are presented as means ± SD. *P < 0.1, **P < 0.01, ***P < 0.001, ****P < 0.0001 (unpaired t test).

**Figure 12. Astrocytic Wnt5a and Wnt7a mRNA up-regulation correlates with blood–spinal cord barrier restoration in amyotrophic lateral sclerosis under chemogenetic activation of astrocytic Gi signaling.**
**(A)** Representative panel showing immunostaining for CLN-5 (red) and COL-IV (green) in WT and *SOD1*[G93A] mice at P38 treated with vehicle or porcupine (PORCN) inhibitor C-59. Breaks of CLN-5 ribbon (red) along the vessels (green) are marked with yellow lines. Scale bars: 20 μm. **(B)** Quantification of CLN-5 breaks length (expressed as % of WT [treated with vehicle] breaks) in WT and *SOD1*[G93A] mice treated with vehicle or C-59 (column with milled pattern). Data are from N = 3. **(C, E)** Representative picture of detection by in situ *hybridization* of (C) Wnt5a and (E) Wnt7a mRNA (green dots) in astrocytes, identified by glial fibrillary acidic protein immunofluorescence staining (red)

but do not have a better disease course and survival (Watanabe-Matsumoto et al, 2018). Our data show that the activation of D(Gi) in astrocytes restores BSCB integrity without ameliorating the burden of multiple disease markers in MN (effectively dissociating the BSCB from other disease manifestations). This suggests that at the initial stages of the disease, BSCB impairment does not necessarily carry pathogenic consequences.

At the same time, our findings are compatible with a possible pathogenic role of BSCB impairment later in disease progression because at this stage, Gi signaling in astrocytes restores the BSCB (once again independently of MN firing), while it reduces the burden of disease markers. However, some limitations apply to later-stage results: Gi signaling activates a number of biological processes in astrocytes, whereas the role of astrocytes may change over time (Ouali Alami et al, 2018). Although it is not possible to draw an unequivocal causal link between BSCB restoration and disease burden, it is still plausible that at this stage, BSCB restoration might be the consequence of the decrease in disease burden.

In conclusion, we demonstrated that BSCB impairment originates as a consequence of early MN dysfunction and can be dissociated from MN disease burden through the selective activation of Gi signaling in astrocytes and the consequent induction of Wnt proteins (in particular Wnt5a/7a). This discovery implies that BSCB disruption does not have a pathogenic role at the early stages of the disease. However, prolonged Gi signaling in astrocytes beneficially affects BSCB function at later stages of the disease by decreasing or delaying MN disease burden.

# Materials and Methods

## Experimental animals

The experiments were approved by the Tierforschungszentrum-Ulm and by the Regierungspräsidium Tübingen (Germany) under license no. 1404. The following strains of transgenic mice were used: *B6SJL-Tg(SOD1\*G93A)1Gur/J* (high-copy, henceforth *SOD1$^{G93A}$*) and *B6.Cg-Tg(SOD1)2Gur/J* (henceforth WT-SOD1) mice were obtained from Jackson laboratories; *B6N.129S6(B6)-Chattm2(cre)Lowl/J* (henceforth *ChAT-Cre*) were a kind gift from Pico Caroni (FMI); *C57BL/6NCrl-FUS(ΔNLS)* (henceforth *FUS$^{(ΔNLS+/−)}$*) were obtained from Luc Dupuis (INSERM; [Scekic-Zahirovic et al, 2016]) and bred locally; *B6SJL/F1-TDP-43(G298S)* (henceforth *TDP-43$^{(G298S)}$*) were a kind gift from Phil Wong (Johns Hopkins School of Medicine [Wiesner et al, 2018]); and *B6.129P2-Tbk1tm1Aki* (henceforth *Tbk1$^{+/−}$*) were a kind gift from Jochen Weishaupt (Ulm University, Ulm; [Brenner et al, 2019]). To generate *SOD1$^{G93A}$/ChAT-Cre* double-transgenic mice, *SOD1$^{G93A+}$* male mice were crossed with *ChAT-Cre$^{+/+}$* female mice, and the double-positive male mice in the F1 were used for the experiments.

All experiments were carried out on male mice because of the known difference in disease progression rate between male and female mice and of the less predictable time course in female mice, starting from the age of P20. Unless otherwise specified, animals were kept in groups of 3–4 mice in a 12-h light–dark cycle and were given ad libitum access to food and water. Mice were checked daily for the appearance of symptoms. Based on power calculations ($α$ = 80% with 5% significance two-sided) for each experimental group or time points, 6–8 animals were processed and analyzed.

## AAV vectors

The following AAV vectors were obtained from Addgene: AAV serotype 8, pAAV(8)-GFAP-GFP, pAAV(8)-GFAP-hM3D(Gq)-Citrine, pAAV(8)-GFAP-hM3D(Gs)-Citrine, and pAAV(8)-GFAP-hM4D(Gi)-Citrine. The plasmids encoding for pAAV(9)-CBA-GFP-2A-floxed-PSAM(L141F,Y115F)GlyR-WPRE and pAAV(9)-pCAG-A7-floxed-PSAM(L141F, Y115F)5HT3-WPRE were a kind gift from Scott Sternson, HHMI-Janelia (Magnus et al, 2011), and were assembled in AAV9 vectors (details of the preparation of the vectors are reported in Supplemental Data 1; Matsushita et al, 1998; Aurnhammer et al, 2011; Hussaini et al, 2013; Jungmann et al, 2017; Commisso et al, 2018).

## Intraspinal injection of AAV

Intraspinal injection of AAV was performed as previously reported (Saxena et al, 2013). Briefly, the spinal cord was accessed by a dorsal laminectomy performed at the T11–T13 level. Injection (1 $μl$) was performed using a pulled glass capillary coupled to a Picospritzer-III apparatus. Using the central dorsal artery as reference, injections were performed at the coordinates y = +0.30; z = −0.45. Further details are reported in Supplemental Data 1.

## Chemogenetics agonist administration

CNO was purchased from Tocris and was administered by i.p. injection at a dose of 5 mg/kg once daily, starting 10 d after the intraspinal injection (unless otherwise specified) for 7 consecutive days. The effector molecule PSEM$^{308}$ was custom synthesized by Apex Scientific and was administered daily at a dose of 5 mg/kg in saline, starting from 10 d after viral injection for 7 consecutive days; because PSEM$^{308}$ is poorly soluble in cold saline, stock solutions were pre-warmed at 40°C for 1 h before injection and used at RT.

---

in WT and *SOD1$^{G93A}$* mice. Nuclei are depicted with DAPI (blue). **(D, F)** Quantification of mRNA levels of (D) Wnt5a and (F) Wnt7a in WT and *SOD1$^{G93A}$* mice, expressed as the amount of dots. **(G, I)** Representative panel showing detection by in situ *hybridization* of (G) Wnt5a and (I) Wnt7a mRNA (green dots) in astrocytes in the inhPSAM/PSEM$^{308}$ chemogenetic experiment. Astrocytes are identified by glial fibrillary acidic protein immunofluorescence staining (red). Nuclei are depicted with DAPI (blue). **(H, J)** Quantification of (H) Wnt5a and (J) Wnt7a mRNA amount in infected and uninfected contralateral ventral horns of the spinal cord of *SOD1$^{G93A}$/ChAT-cre* mice injected with AAV9 inhPSAM/PSEM$^{308}$. Data are from N = 4. **(K, M)** Representative panel showing detection by in situ *hybridization* of (K) Wnt5a and (M) Wnt7a mRNA (green dots) in astrocytes in the D(Gi)/CNO chemogenetic experiment. Nuclei are depicted with DAPI (blue). **(L, N)** Quantification of (L) Wnt5a and (N) Wnt7a mRNA amount in infected and uninfected contralateral ventral horns of the spinal cord of *SOD1$^{G93A}$/ChAT-cre* mice subjected to D(Gi)/CNO chemogenetic treatment. Data are from N = 4. Scale bars: 5 $μm$. Data information: in (B, D, F, H, J, L, N), data are presented as means ± SD. **(B, D, F, H, J, L, N)** *$P < 0.1$, **$P < 0.01$, ***$P < 0.001$, ****$P < 0.0001$ (one-way ANOVA with Bonferroni correction for multiple comparisons [B] and unpaired *t* test [D, F, H, J, L, N]).

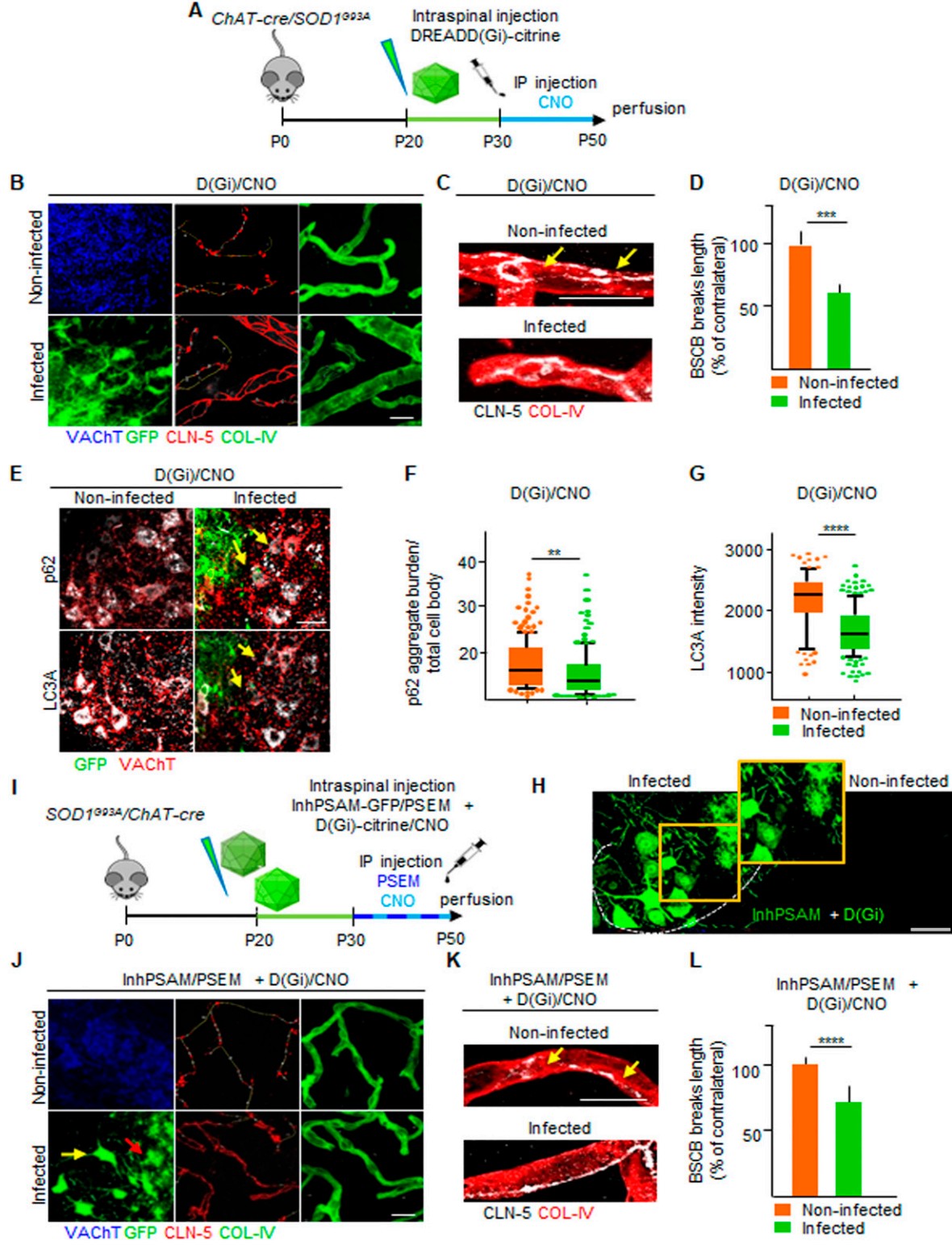

**Figure 13. Chemogenetic activation of astrocytic D(Gi) at later amyotrophic lateral sclerosis stage ameliorates microvessel integrity and MN disease markers.**
**(A)** Experimental design for the injection of AAV8 encoding D(Gi) in *SOD1^G93A^/ChAT-cre* mice, at P20 and treated with the agonist CNO from P30 until P50. **(B)** Representative panel displaying the effect of the prolonged and late activation of astrocytic (DGi) on CLN-5 distribution (red) along the vessels (green). MNs are identified by VAChT immunostaining (blue), and infected astrocytes are stained with GFP (green). CLN-5 breaks are depicted in yellow lines. **(C)** High magnification of CLN-5 organization (white) along COL-IV⁺ vessels (red). Yellow arrows indicate CLN-5 interruptions along the ribbon in the noninfected ventral horn of *SOD1^G93A^/ChAT-cre* mice subjected to D(Gi) late and prolonged activation. Scale bars: 10 μm. **(D)** Quantification of blood–spinal cord barrier (BSCB) disruption, expressed as % of contralateral

### C59 administration

The C59 inhibitor of porcupine (PORCN) was suspended in PBS + 5% Tween 80 + 5% PEG-400 and was administered to SOD1[G93A] and WT mice at the age of 20–22 d, via oral gavage (10 mg/kg) for 14 d. Vehicle (composed of PBS, Tween 80, and PEG-400) was administered to control SOD1[G93A] and WT mice.

### Western blot

For the preparation of whole spinal cord protein extract, the lumbar spinal cord was dissected after mouse euthanasia by cervical dislocation and snap frozen on dry ice as previously reported (Olde Heuvel et al, 2019). Briefly, lumbar spinal cord tissue was dissected and homogenized in a complete radioimmunoprecipitation assay (RIPA) buffer, containing protease and phosphatase inhibitors. Samples were then subjected to SDS/PAGE gel electrophoresis, transferred to a polyvinylidene difluoride (PVDF) or nitrocellulose membrane, and subsequently incubated overnight at 4°C with the following primary antibodies: total anti-TDP 43 rabbit (10782-2-AP; 1:2,000 buffered in 1% BSA in PBS containing 0.05% Tween 20; Proteintech), anti-FUS rabbit (1:2,000 buffered in 1% BSA in PBS containing 0.05% Tween 20; Bethyl Laboratories), and anti-SOD1 rabbit (Prestige 001401; 1:1,000 buffered in 1% BSA; in PBS containing 0.05% Tween 20; Sigma-Aldrich). After washing steps in PBS enriched with 0.05% Tween 20, the membranes (nitrocellulose membrane for SOD1 and PVDF for TDP43 and FUS) were subsequently incubated for 1 h at RT with goat anti-rabbit IgG-HRP-conjugated secondary antibody (1:10,000 in PBS-Tween 20; Bio-Rad). After appropriate washing steps, the membranes were further treated with Western ECL-immunodetection buffer (Bio-Rad) and acquired using Image Lab 5.0. ThermoScientific PageRuler Plus Prestained was used as a protein ladder. Samples were corrected for background, and densitometry analysis was performed using Image Lab software5.0. To control variability in loading individual samples, the signal intensity of protein bands was normalized to housekeeping GAPDH protein.

### Histology and immunostaining

The spinal cord was prepared for histology and immunostaining as previously reported (Saxena et al, 2013; Ouali Alami et al, 2018). Briefly, after perfusion, fixation with 4% PFA, and cryoprotection in 30% sucrose, 40-$\mu$m-thick sections were subject to antigen retrieval (whenever indicated) and immunostained using a free-floating protocol. Details of the immunostaining procedure are reported in Supplemental Data 1, and a list of the antibodies and other reagents used is reported in Table S1.

### Claudin-5 breaks quantification

The loss of CLN-5 ribbon is defined by a lack of CLN-5 immunostaining under specific threshold by taking as reference the capillary wall stained with collagen-IV. The length breaks are measured and traced (in yellow) with ImageJ software. The quantification is performed by relating the total length of the single breaks and the total length of the vessels, outlined by collagen-IV, and expressed as % of breaks on the total vessels. Images are acquired as single tile scans, covering the ventral horn of the spinal cord, with specific Z-stack at the confocal microscope and analyzed as collapsed pictures.

### Single-molecule in situ mRNA hybridization

Detection of mRNA in situ together with co-immunostaining was performed as previously reported (Olde Heuvel et al, 2019) and according to the manufacturer's recommendation (Acd Bio). Details of the procedure are reported in Supplemental Data 1, and details of the probes used are reported in Table S1.

### Confocal imaging and image analysis

Confocal images were acquired as previously reported (Ouali Alami et al, 2018). Details of image acquisition are reported in Supplemental Data 1.

The quantification of disease burden markers (misfolded SOD1, LC3A, KDEL, and p62) was performed as previously reported (Saxena et al, 2013; Ouali Alami et al, 2018). Detailed procedures are reported in Supplemental Data 1.

For the quantification of the structural disruption of the BSCB, we considered a ROI in the ventral spinal cord. We traced the total length of vessels, covered by collagen-IV staining, in the ROI and the length of vascular segments in which the claudin-5 ribbon appeared disrupted or fragmented ("gaps" or "breaks," depicted in yellow lines) and computed the ratio between the cumulative gap length and the total length of the vessels, as previously reported (Ouali Alami et al, 2018).

breaks length. **(E)** Representative pictures showing MN markers p62 and LC3A immunofluorescence staining (gray) in infected and uninfected contralateral ventral horn of *SOD1[G93A]/ChAT-cre* spinal cord sections. MNs are detected by VAChT immunostaining (red) and infected astrocytes by GFP (green). **(F, G)** Quantification of (F) p62 aggregates per cell body and (G) LC3A intensity in MN surrounding infected astrocytes and in MN located in the uninfected contralateral ventral horn. The quantifications are represented by the box-and-whisker plot; 10–90 percentile is considered. Data are from N = 3 mice. **(I)** Experimental design for multiplexed chemogenetic injection of AAV9 encoding inhPSAM + D(Gi) in *SOD1[G93A]/ChAT-cre* mice, at P20 and treated with respective ligands PSEM[308] and CNO from P30 until P50. **(H)** Expression pattern of MNs activated by inhPSAM (green) and astrocytes activated by D(Gi). Dotted lines delineate the contour of the gray and white matter in the ventral horns of the spinal cord. The insert highlights infected MNs (green) surrounded by infected astrocytes (green). Scale bar: 50 $\mu$m. **(J)** Representative panel displaying the effect of the prolonged and late activation of multiplexed chemogenetic inhPSAM/PSEM[308] + D(Gi)/CNO experiments on the BSCB grade of disruption in *SOD1[G93A]/ChAT-cre* mice. MNs in the noninfected horn are identified by VAChT immunostaining (blue), whereas infected MNs and astrocytes are stained with GFP (green). CLN-5 breaks are depicted in yellow lines along the ribbon (red). **(K)** High magnification of CLN-5 organization (white) along COL-IV[+] vessels (red). Yellow arrows indicate CLN-5 interruptions along the ribbon in the noninfected ventral horn of *SOD1[G93A]/ChAT-cre* mice subjected to the late and prolonged multiplexed chemogenetic experiments. Scale bars: 10 $\mu$m. **(L)** Quantification of breaks length in the BSCB of *SOD1[G93A]/ChAT-cre* mice subjected to the late and prolonged multiplexed chemogenic experiments, expressed as % of contralateral. Data are from N = 3 mice. Scale bars: 20 $\mu$m. Data information: in (D, F, G, L), data are presented as means ± SD. **P < 0.01, ***P < 0.001, ****P < 0.0001 (unpaired t test).

For the quantification of the astrocytic end-feet coverage of spinal cord vessels, blood vessels were first identified using collagen-IV staining, and a ROI corresponding to the vessels was manually traced; the surface of the vessel occupied by GFP-positive processes was then quantified, and a ratio between the GFP$^+$ area and the total vessel area was considered. The same procedure was used for the quantification of AQP4 around the microvessels. For quantitative analysis, a minimum of 8–10 artifact-free sections per mouse was analyzed, and from each 16–20 vessel, stretches were taken into consideration.

## Statistical analysis

One-way ANOVA with Bonferroni correction for multiple comparisons was applied for the comparison of multiple groups. For the comparison of multiple groups with various conditions, two-way ANOVA and nonparametric statistical analysis were applied. The unpaired $t$ test was used to determine if the difference between two groups is significant. Statistical analysis was performed with Prism software (GraphPad6 and 8). All values were expressed as mean ± SD unless otherwise indicated. Statistical significance was set at $P < 0.05$ before multiple comparison correction.

# Supplementary Information

# Acknowledgements

F Roselli is supported by the Synapsis Foundation, the Thierry Latran Foundation (projects "Trials" and "Hypothals"), the Radala Foundation, the Deutsche Forschungsgemeinschaft (DFG, German Research Foundation)—Project ID 251293561—Collaborative Research Center 1149 and with the individual grant nos. 431995586 (RO-5004/8-1) and 443642953 (RO5004/9-1), the Cellular and Molecular Mechanisms in Aging (CEMMA) Research Training Group, and Bundesministerium für Bildung und Forschung (BMBF) (FKZ 01EW1705A, as member of the ERANET-NEURON consortium "MICRONET"). N Ouali Alami and B Commisso are members of the International Graduate School in Molecular Medicine, Ulm University. N Ouali Alami is currently supported by the Baustein grant of the Ulm University Medical Faculty. L Tang is supported by the China Scholarship Council. D Bayer is supported by the Cellular and Molecular Mechanisms in Aging (CEMMA) Research Training Group. The authors are grateful to Dr. Memet Sacma (Molecular Medicine Department, University of Ulm) for providing the CD31/PECAM-1 antibody, Clara Bruno (Neurology Department, University of Ulm) for providing the *Tbk1* spinal cord lysates, and D Wiesner (Neurology Department, University of Ulm) for providing access to the *TDP-43*$^{G298S}$ and FUS ALS mouse lines. The authors wish to thank Dr. Christopher Geekie for carefully proofreading the manuscript and Prof. Frank Kirchhoff (Virology department, University of Ulm) for the use of the Zeiss LSM710 confocal microscope, Prof. Anita Ignatius (Trauma surgery research and biomechanics department, University of Ulm) for the use of the histology facility, and Thomas Lenk, Tanja Wipp, and Florian olde Heuvel for the dedicated technical support.

## Author Contribution

N Ouali Alami: conceptualization, data curation, formal analysis, investigation, visualization, methodology, and writing—original draft, review, and editing.

L Tang: data curation, formal analysis, investigation, methodology, and writing—original draft, review, and editing.

D Wiesner: resources, investigation, and writing—review and editing.

B Commisso: investigation and methodology.

D Bayer: resources and methodology.

J Weishaupt: resources.

L Dupuis: resources.

P Wong: resources.

B Baumann: resources.

T Wirth: resources and writing—original draft, review, and editing.

TM Boeckers: resources, data curation, formal analysis, and writing—original draft, review, and editing.

D Yilmazer-Hanke: data curation, supervision, and writing—review and editing.

A Ludolph: conceptualization, resources, and writing—original draft, review, and editing.

F Roselli: conceptualization, resources, formal analysis, supervision, funding acquisition, validation, project administration, and writing—original draft, review, and editing.

## Conflict of Interest Statement

The authors declare that they have no conflict of interest.

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
