## [Reviewer comments · Life Science Alliance]

Life Science Alliance

Multiplexed chemogenetics in astrocytes and motoneurons restore Blood-Spinal Cord-Barrier in ALS

Najwa Ouali Alami, linyun Tang, Diana Wiesner, Barbara Comisso, David Bayer, Jochen Weishaupt, Luc Dupuis, Philip Wong, Bernd Baumann, Thomas Wirth, Tobias Boeckers, Deniz Yilmazer-Hanke, Albert Ludolph, and Francesco Roselli

DOI: <https://doi.org/10.26508/lsa.201900571>

Corresponding author(s): Francesco Roselli, Ulm University and Najwa Ouali Alami, Ulm University

Review Timeline:

Submission Date:	2019-10-03
Editorial Decision:	2019-11-11
Revision Received:	2020-07-28
Editorial Decision:	2020-08-14
Revision Received:	2020-08-24
Accepted:	2020-08-31

Scientific Editor: Shachi Bhatt

Transaction Report:

November 11, 2019

Re: Life Science Alliance manuscript #LSA-2019-00571-T

Prof. Francesco Roselli
Ulm University
Neurology
Helmholtzstrasse 8/2
Ulm 89081
Germany

Dear Dr. Roselli,

Thank you for submitting your manuscript entitled "Chemogenetics dissociates Blood-Brain-Barrier disruption from disease burden in ALS through Wnt5a/7a" to Life Science Alliance. The manuscript was assessed by expert reviewers, whose comments are appended to this letter.

As you will see, while the reviewers appreciate the aim of your work, they also think that your conclusions are not sufficiently supported. They provide constructive input on how to add more support and we would like to invite you to submit a revised version of your manuscript to us.

Importantly, please provide functional correlates for barrier disruption occurring independently of motoneuron loss and more insight into barrier disruption (reviewer #1 and #2), and address all requests pertaining to data quality and missing controls (all reviewers). Please also address the concern of false positives for the blood vessel stainings (reviewer #2). It is not mandatory to analyze later disease stages as suggested by reviewer #2 and the request of reviewer #3 regarding the Wnt pathway can get addressed in discussion.

Please also note that reviewer #1 pointed out upon further discussing the manuscript that a more intuitive model (current EV4) that does not require reading the manuscript, should get provided. You may want to add numbers to guide the reader through the model figure.

When submitting the revision, please include a letter addressing the reviewers' comments point by

point.

Thank you for this interesting contribution to Life Science Alliance. We are looking forward to receiving your revised manuscript.

Sincerely,

B. MANUSCRIPT ORGANIZATION AND FORMATTING:

Reviewer #1 (Comments to the Authors (Required)):

This work builds on earlier work published in the EMBO Journal (2018) that has shown disruption of the blood-spinal cord barrier (BSCB) in mouse models of ALS and accelerated leakage as shown by staining for immunoglobulins. Here, the team aimed to determine whether the BSCB impairment is the cause or consequence of motoneuron dysfunction and whether restoration of BSCB may be directly beneficial.

I generally like this study and was not aware of the massive breakdown of the BSCB in ALS mouse models. The study (as mentioned) refers back to an earlier study in the EMBO J, using similar analyses tool. In particular it is being reported that in the ALS models there are breaks in the tight junction protein claudin-5 which correlate with the extravasation of immunoglobulin and fibrinogen. In their new study which uses chemogenetics to address the cause and consequence question (using quite short time frames) the only readout being used for determining restored function is claudin-5 breaks.

(1) What is needed, is a functional readout, i.e. IgG extravasation.

(2) It is not clear what a break actually means. Is there a thinning of claudin-5 ribbon or is claudin-5 in the TJs completely gone (difficult to see with the resolution of the images). What is needed is a better way to define what is actually meant by a 'break'. The team should use in addition antibodies for occludin and possibly adherens junction proteins to better convey how the BSCB is actually impaired. Is there the option of running a biochemical analysis (is claudin-5 reduced or internalised)?

(3) In their figures a single WT column is displayed for the different ages. I don't think this is appropriate as in the EMBO paper some WT values vary significantly with age.

(4) Displaying breakage as %age of WT is also not informative. I would like to know what the %length is of claudin-5 breaks over the entire capillary length. Is it 10%, 1% or 0.001%?

(5) Is there any correlation with neurological symptoms?

(6) Regarding the different transgenic strains, please provide the expression levels relative to WT in the SC and at the time when the data were obtained.

(7) Figure 2: It would be helpful to display an example of the human pathology (CLD5-breaks) next to the mouse images.

(8) Do the authors (besides from a signal pathway point of view) have any idea how the break is restored? What does this imply/mean at a cellular level?

(9) Some of the data are not convincing, see Fig 3J (z stacks?) or EV1. The quality of other figures (e.g. Figure 4) is very poor (in fact hard to conclude a lot based on the images) and a biochemical read-out would substantiate the claims.

Minor comments:

(1) It seems to me that the first paragraph about the disruption of the BBB or BSCB in brain diseases is a bit oversimplified. Further down on this page it should be specified whether the in vitro models of the BSCB are human or murine. It is also unclear what these cells are modeling - a specific disease? There is further reference of pericytes but the study did not address the role of

this cell-type in their analysis.
(2) Please use page numbers.

Reviewer #2 (Comments to the Authors (Required)):

In the present study of Alami et al. entitled "Chemogenetics dissociates Blood-Brain-Barrier disruption from disease burden in ALS through Wnt5a/7a" the authors study the timely origin of Blood-Spinal Cord-Barrier (BSCB) disruption in different mouse models of ALS. Hereby, they show that BSCB can be uncoupled from motor neuron degeneration. Instead, BSCB is associated with early excitatory MN dysfunction. Furthermore, they show that BSCB can be restored by targeting D(Gq), but not D(Gi) G-protein coupled signaling in astrocytes. Mechanistically, Wnt5a and Wnt7a secreted by astrocytes are important to maintain the BSCB. The findings of these studies are interesting and novel as they shed light onto the mechanistic role of astrocytes and the endothelium in contribution to ALS phenotype.

Though the results are encouraging and of important impact for the field, there are some limitations of the study that should be addressed before publication.

Major comments:

- The authors use different mouse models of ALS that, according to the literature, present severe or no motor neuron loss. However, the data shown in Figure 1 is not clearly correlated to the state of motor neuron loss or cell death in the respective mouse model. Even though the data presented has been related to already published findings, the authors should provide stainings of spinal cord motor neurons and/or cell death (TUNEL, cCasp-3) in combination with the Claudin-5/vessel staining to transmit their hypothesis that Claudin-5 breaks happen independently of motor neuron cell death, more clearly.
- Throughout the study, Collagen IV has been used as blood vessel marker. The disadvantage of this marker though is that it only stains the endothelial cell basal membrane. It could be that vessel remodeling takes place during ALS and blood vessels regress. In that case, the Collagen IV matrix could still be detected, though there are no blood vessels anymore. This might lead to a false positive assumption of Claudin-5 breaks resulting from the complete absence of blood vessels.
- In line with the above comment, is it known that only vessel permeability is affected, or is the blood vessel density per se also affected in ALS?
- To complement the Claudin-5 analysis, the authors should use a functional assay for the BSCB and inject a fluorescent tracer to analyze vessel permeability. This analysis should be done in both basal disease conditions as well as in the rescue experiments targeting astrocytes and MN excitability. Alternative, staining for fibrinogen and/or immunoglobulin would be important to really demonstrate extravasation and therefore compromised barrier function.
- Is only the localization affected or also the total protein levels of Claudin-5? Is the mechanism specific for Claudin-5, or are other junctional proteins affected as well (e.g. VE-Cadherin)?
- The efficiency of MN excitation/inhibition after viral delivery of actPSAM/inhPSAM should be demonstrated to show that the model system is indeed working as expected.

- The authors claim that increasing Gi signaling in astrocytes also restores later disease stages. However, at least from the data provided, this conclusion is not fully supported. The authors show that the first, mild Claudin-5 breaks occur at P20 in the SOD mutant, whereas the peak of BSCB starts at around P50. Instead of restoring Gi signaling between P30-P50, the authors should treat the mice at later disease stages (e.g. P50) and analyze the mice at P80.
- Is astrocytic endfeet coverage affected in the different models of ALS used?
- How can the authors relate the restoration of Claudin 5 that occurs with the astrocytic Gi/Gq with the recovery of ALS symptoms? It could be that "protection of the BSCB" is a parallel event and not astrocytic dependent. It could be that astrocytes are also signaling to neurons, microglia etc, which could then lead to the effect on neurons. It is recommended to further interpret their data with other potential mechanisms, in the discussion of the paper.

Minor comments:

- The authors study different disease time points defined as 'presymptomatic, early symptomatic, fully symptomatic stages'. Experimental details should be provided on how these categories have been defined.
- It is sometimes difficult to follow the reasoning behind certain experiments, especially the part about G-protein couples signaling pathways in astrocytes. Please provide a better introduction of the scientific background. Also, many abbreviations, such as 'Gi, Gs or Gq' are not introduced, making it difficult to read the abstract and introduction.
- The manuscript should be edited/revised by a professional/native speaker due to some minor, but repetitive language mistakes.
- Please provide high resolution figures as well as better representative images.
- Figure 2 could be nicely complemented with a graph showing infection efficiency of the MN population.
- Show representative images of LC3A staining (autophagy marker) as complementary to the presented quantification graph (Figure 4).
- In Figure EV1 the authors show astrocytic endfeet coverage, however they do not use AQP4 as a marker for the endfeet.
- In Figure EV2 there is no GFAP/AQP4 staining showing astrocytes in SOD1 mice. Do the authors see a loss in the number of astrocytes or just activity alterations in this model?
- Figure 6: Wnt5a/Wnt7a staining/ISH is not visible in the images they show. Also, based on what they describe, Wnt staining should also be detected in MNs. Would combine Wnt ISH with MN specific staining to further demonstrate this. Maybe also perform westernblot to show the expression.
- Figure4- Insets to see the misfolded SOD1-the image quality is very poor. What do the grey cells

indicate-which marker is that?

- What is the molecular mechanism of suppressed Wnt5a and 7a release due to MN dysfunction? It would be nice if the authors could further elaborate that point in the discussion

Reviewer #3 (Comments to the Authors (Required)):

The manuscript by Alami et al. provides interesting data describing the relationship of motoneuron (MN) activity and blood-spinal Cord-Barrier (BSCB) under the pathological conditions in a mouse model of Amyotrophic Lateral Sclerosis (ALS). Specifically, the authors show that BSCB impairment was independent of MN loss in different models of ALS, namely in SOD1(G93A), FUS(Δ NLS), TDP43(G298S) and TBK1+/- ALS mouse models. Interestingly, in SOD1G93A ALS mice the BSCB was restored by increased MN firing, but worsened by MN inactivation, using a chemogenetic approach. The authors could further demonstrate that specific modulation of GPCR signaling in astrocytes (ACs) by clozapine-N-oxide (CNO)-inducible activation of D(Gs), D(Gi) and D(Gq) signaling downstream of the GFAP promoter after AAV infection.

The authors show that D(Gi) and D(Gq) were able to restore BSCP leakage in SOD1(G93A) mice, whereas D(Gs) had no significant effect. The BSCP restoration had no beneficial effect on MN disease markers, which was interpreted by the authors to be indicative of independent processes. Finally, Alami et al. provide evidence for a decreased expression of Wnt7a and Wnt5a in SOD1(G93A), which however could strongly be enhanced by Gi signaling, but further decreased by MN inactivation.

Together, the presented data this manuscript demonstrate that in ALS the BSCB is indeed impaired in various mouse models of the disease, and that this is related to MN firing as well as AC GPCR signaling.

The data are solid and the presentation of the data is very good. However, to augment the scientific merit of the manuscript, some issues require the authors attention:

Major points

One major concern of the reviewer regards the Wnt pathway activity in the vasculature of the spinal cord. The authors assume that the down-regulation of Wnt5a and Wnt7a in ACs leads to reduced Wnt activation in endothelial cells (ECs). However, this has to be supported by experimental evidence. The authors should stain for Lef1 and/or isolate vessels to monitor expression of Wnt target genes like Axin2, Nkd1 or Sox17 in ECs. Wnt5a might also inhibit the beta-catenin-driven Wnt pathway. The authors should take this also into consideration.

Does DREADD activation that leads to Wnt5a and Wnt7a expression also augment Wnt/beta-catenin signaling in ECs?

Additional points are raised below:

Figure 1B, C, D: Vessels of the considerably longer aged (P150-P270) WT mice of the FUS(Δ NLS), TDP43(G298S) and TBK1+/- ALS mouse models also show stretches devoid of Cldn5. Is this an ageing phenotype that the authors did not observe in the younger (P20) SOD1(G93A) mouse model?

Figure 6 C-N: The staining and hence the quantification of Wnt5a and Wnt7a is not convincing. Other, quantitative methods such as western blotting and qRT-PCR would be ideal to complement the finding.

Figure EV1 and EV2: The staining of GFAP, and in particular of Aqp4, do not convincingly show vessel coverage, leaving doubt with regard to the quantification. Please provide improved images.

Minor points

Page 6, line 9: Which software has been used for vessel and leakage quantification?

Page 17, line 19: Please change "..., and it sufficient to induce a BBB-like phenotype in endothelial cells [4] and on Wnt 7a, a critical mediator of the establishment and maintenance of the BBB phenotype [54, 12]." into "..., and is sufficient to induce a BBB-like phenotype in endothelial cells [4] and on Wnt 7a, a critical mediator of the establishment and maintenance of the BBB phenotype [54, 12]."

Overall the manuscript by Alami et al. describes interesting findings. Hence the reviewer recommends the manuscript for publication after major revision.

Thank you for considering this revised version of the manuscript entitled "Chemogenetics dissociates Blood-Brain-Barrier disruption from disease burden in ALS through Wnt5a/7a" for publication in Life Science Alliance. The revision was delayed by a series of unforeseen events, including equipment failure, delayed the supply chain and by the university-wide shut-down due to the Coronavirus-19 crisis that prevented any significant activity for several weeks.

We have clarified all the main points raised by the reviewers and underscored by the editorial board. The additional evidence requested has required extensive experiments and the current manuscript has now twice as many figures as the original manuscript; we are available to follow the guidance of the editorial board should the number of figures prove too large. We provide below a point-by-point rebuttal. Changes in the manuscript are highlighted in blue.

We look forward to hearing from you, hopefully for a positive feedback.

Editorial board note:

Importantly, please provide functional correlates for barrier disruption occurring independently of motoneuron loss and more insight into barrier disruption (reviewer #1 and #2), and address all requests pertaining to data quality and missing controls (all reviewers). Please also address the concern of false positives for the blood vessel stainings (reviewer #2). It is not mandatory to analyze later disease stages as suggested by reviewer #2 and the request of reviewer #3 regarding the Wnt pathway can get addressed in discussion.

Please also note that reviewer #1 pointed out upon further discussing the manuscript that a more intuitive model (current EV4) that does not require reading the manuscript, should be provided. You may want to add numbers to guide the reader through the model figure.

> We have now replaced the explicative diagram (now Supplementary Figure 5) with a new, streamlined one, using numbers to guide the reader through. We have also modified the title in order to better represent the content of the paper.

Reviewer #1 (Comments to the Authors (Required)):

This work builds on earlier work published in the EMBO Journal (2018) that has shown disruption of the blood-spinal cord barrier (BCSB) in mouse models of ALS and accelerated leakage as shown by staining for immunoglobulins. Here, the team aimed to determine whether the BSCB impairment is the cause or consequence of motoneuron dysfunction and whether restoration of BSCB may be directly beneficial.

I generally like this study and was not aware of the massive breakdown of the BSCB in ALS mouse models. The study (as mentioned) refers back to an earlier study in the EMBO J, using similar analyses tool. In particular it is being reported that in the ALS models there are breaks in the tight junction protein claudin-5 which correlate with the extravasation of immunoglobulin and fibrinogen. In their new study which uses chemogenetics to address the cause and consequence question (using quite short time frames) the only readout being used for determining restored function is claudin-5 breaks.

(1) What is needed, is a functional readout, i.e. IgG extravasation.

>We have now integrated the characterization of the BSCB dysfunction with the measure of albumin extravasation by immunostaining of spinal cord sections. Albumin immunoreactivity was completely absent in WT samples (with the exception of some residual albumin localized inside the vessels) but was significantly increased in the SOD1^{G93A} line as well as in the TDP-43^{G298S}, FUS^{ΔNLS/+} and Tbk1^{+/-} mouse lines. Notably, the extent of intraparenchymal Albumin increased with age in the ALS mice but not in their WT littermates. This finding further supports the concept of functional and structural BSCB disruption in multiple ALS mouse models. We have provided the additional information in Figure 2.

(2) It is not clear what a break actually means. Is there a thinning of claudin-5 ribbon or is claudin-5 in the TJs completely gone (difficult to see with the resolution of the images). What is needed is a better way to define what is actually meant by a 'break'. The team should use in addition antibodies for occludin and possibly adherens junction proteins to better convey how the BSCB is actually impaired. Is there the option of running a biochemical analysis (is claudin-5 reduced or internalised)?

>"Breaks" are intended as focal areas of impaired blood-spinal cord-barrier. In high-magnification confocal images the claudin-5 ribbon appears continuous in WT animals, but displays segments of fragmentation in the ALS mice. At high magnification, in the areas of fragmentation, isolated clusters of claudin-5 can be detected. We now show that a similar phenotype appears for zonula occludens-1 (ZO-1): in spinal cord sections from WT animals, ZO-1 appears to define a continuous ribbon inside the vessel lining, however in the ALS mice the ribbon displays focal disruptions in which only small ZO-1 clusters are seen. The timecourse of ZO-1 disruption is consistent with that of claudin-5 in the SOD1^{G93A} mouse as well as in TDP-43^{G298S}, FUS^{ΔNLS/+} and Tbk1^{+/-} mice. These alterations have focal distribution, i.e. the vessel may display the continuous ribbons of claudin-5 and ZO-1 before and after the "break". Because of the focal nature of the alteration in BSCB and because of their spatial restriction to the ventral horn, biochemical analysis is not possible. We have added the ZO-1 data in the new Figure 2.

(3) In their figures a single WT column is displayed for the different ages. I don't think this is appropriate as in the EMBO paper some WT values vary significantly with age.

>We have now provided representative figures and break % for each timepoint of the WT animals. WT animals display a degree of claudin-5 breaks ranging between 10 and 30% at the earliest timepoints; factors that contribute to this value include some intrinsic inefficiency of immunostaining procedure and line-specific features (which equally affect WT and ALS mice and therefore are not the cause of the differences observed). We detected a marked

increase in break length in WT mice older than 500 days, in agreement with the progressive decline in Blood-Brain-Barrier with aging. We have added these data to Figure 1.

(4) Displaying breakage as %age of WT is also not informative. I would like to know what the %length is of claudin-5 breaks over the entire capillary length. Is it 10%, 1% or 0.001%?

>We have revised the display of figure 1 to provide the % of breaks for each timepoint independently. We show that in WT animals the % of breaks is approximately between 15 and 30% for animals up to 450 days of age, with a substantial increase observed only in mice older than 500 days. Data are shown in Figure 1

(5) Is there any correlation with neurological symptoms?

>In the case of the high-copy SOD1^{G93A} overt neurological symptoms (stage 1 of the clinical score or peak body weight) usually appear at about P70-P80, as extensively reported in the abundant literature regarding this mouse line (e.g., Boillee et al., 2006; Ouali Alami et al., 2018). The earliest signs of BSCB disruption are observed already at P20, when accumulation of misfolded SOD1 is detected in vulnerable MN (Saxena et al., 2013) but no sign of neurological impairment or muscular denervation is observed (Pun et al., 2006). Nevertheless, alteration of MN excitability is already detected at about P45 (Martinez-Silva et al., 2018). Therefore, the disruption of the BSCB appears as early as the earliest markers of ALS-related pathobiochemistry and pathophysiology and before over neurological dysfunction. For the FUS^{ANLS/+}, behavioural abnormalities have not been reported by the original developers of the mouse line (Scekic-Zahirovic et al., 2017) and by us (Wiesner et al., 2018) before the age of 10 months (300 days); at the age of P150, when a substantial disruption of the BSCB is already detectable, FUS^{ANLS/+} display a comparable grip strength, body weight curves and accelerating rotarod performance as WT littermates (Scekic-Zahirovic et al., 2017); modest worsening of the motor behavior is observed at extreme timepoints (22 months vs 10 months. Thus, once again BSCB disruption appears before overt neurological impairment. TDP-43^{G298S} mice display a modest neurological impairment in motor behaviour at the age of P80, which remains stable till P300, when a progressive worsening is observed (Wiesner et al., 2018). Also in this case, BSCB disruption is already evident when the earliest neurological dysfunctions are detected and worsen with time. In the case of the Tbk1^{+/-} mice, no neurological impairment of any sort has been described so far (these mice also have normal life span; Brenner et al., 2018), yet a substantial disruption of the BSCB is detected at P270. Thus, for these four lines the evidences suggest that BSCB disruption appears at the time of the very first neurological symptoms or before, and becomes more severe in correlation with the worsening of neurological status. We have now concisely underscored in the results chapter the neurological status for each mouse line at the time of BSCB evaluation and we have underscored again this matter in the discussion.

(6) Regarding the different transgenic strains, please provide the expression levels relative to WT in the SC and at the time when the data were obtained.

>We now provide as supplementary information the quantification of the expression of human SOD1, mutant TDP-43 and mutant FUS in spinal cord homogenate from the respective mouse lines, compared to their wt littermates. Transgenic mouse lines expressing mutant SOD1 and TDP-43 displayed higher level of SOD1 and TDP-43, respectively, than wt littermates, in agreement with the genetic engineering strategy used to develop them, the

mutant FUS levels were comparable to WT, since this line has been obtained with a knock-in strategy. The levels of mutant proteins do not change over time, in agreement with previous characterizations of the mouse lines employed (e.g., Ouali Alami et al., 2018; Scekcic-Zahirovic et al., 2017). For the TBK1^{+/-} line, we demonstrate that the levels of TBK1 are decrease to 50-60% of WT animals at all timepoints, as expected in an heterozygous KO and in agreement with previous characterization of the mouse line (Brenner et al., 2018). WBs are displayed in Figure S1

(7) Figure 2: It would be helpful to display an example of the human pathology (CLD5-breaks) next to the mouse images.

>We have tried to obtain meaningful CLN-5 immunostaining on spinal cord obtained from two ALS patients and 2 healthy subjects. To date, despite repeated attempts, we have been unable to produce stainings of sufficient quality; this is due to intrinsic complexity of immunostaining tissue that has been formalin-fixed for weeks or months and that, at the moment of fixation, had already undergone substantial post-mortem alterations. These issues are well recognized in the field of human pathology and are not an isolated problem of our study. We agree that having human samples would further strengthen our findings, but we have to conclude that the availability of such data falls beyond the time scope of the present revision.

(8) Do the authors (besides from a signal pathway point of view) have any idea how the break is restored? What does this imply/mean at a cellular level?

> Components of the Tight-Junctions (TJ), in particular occludin and claudin-5, have been reported to undergo cycling between the surface of endothelial cells and endocytic compartments through a process mediated by caveolins (Stamatovic et al., 2009) and set in motion by cytokines and chemokines, such as CCL2. The removal of claudin-5 from the plasma membrane is controlled by a number of phosphorylation events (involving serine and tyrosine kinases, Stamatovic et al., 2003, 2006). After being endocytosed, claudin-5 and occludin do not appear to undergo rapid degradation, but are intracellularly stored to be trafficked back to the cell surface upon the resolution of the inflammatory stimulus. Of note, we observe that in the damaged sections of the claudin-5 ribbon, the protein itself is not completely absent but it is rather localized in round clusters. It can be therefore speculated that the removal of claudin-5 from the TJs may be due to endocytosis and that the protein, together with other TJ proteins, may be returned back to the cell surface. The breaks would then appear because of the prolonged intracellular sequestration rather than as consequence of the permanent loss of the protein content. In addition, Wnt signaling has been reported to upregulate the transcription of several tight junction proteins, including VE-cadherin and claudin-5 itself (Roudnicky et al., 2020). Thus, it is conceivable that restoration of BSCB integrity may involve transcriptional mechanisms as well as regulation of tight junction protein trafficking. We have now addressed this possible mechanism in the Discussion section.

(9) Some of the data are not convincing, see Fig 3J (z stacks?) or EV1. The quality of other figures (e.g. Figure 4) is very poor (in fact hard to conclude a lot based on the images) and a biochemical read-out would substantiate the claims.

> *We have now extensively changed the display settings and the representative images for most of the figures of the manuscript. Because of the focal nature of the BBB disruption in terms of anatomy (ventral horn) and of the comparative amount of proteins derived from blood vessels in a whole-spinal cord homogenate, substantial dilutional effects may render biochemical assays less sensitive than immunohistological approaches and would not offer a suitable spatial resolution.*

Minor comments:

(1) It seems to me that the first paragraph about the disruption of the BBB or BSCB in brain diseases is a bit oversimplified. Further down on this page it should be specified whether the in vitro models of the BSCB are human or murine. It is also unclear what these cells are modeling - a specific disease? There is further reference of pericytes but the study did not address the role of this cell-type in their analysis.

> *We have removed the first paragraph and focused the introduction on ALS. We have now clarified that the in vitro model (Meisters et al., 2015) refers to primary spinal cord endothelial cells obtained from SOD1^{G93A} mice or immortalized murine endothelial cells expressing either WT or mutant (G93A) SOD1. In order to streamline the introduction, we have removed the reference to the possible role of pericytes in the disruption of the BSCB in ALS.*

(2) Please use page numbers.

> *We thank the reviewers for the suggestion. We have amended the manuscript by adding the page numbers.*

Reviewer #2:

In the present study of Alami et al. entitled "Chemogenetics dissociates Blood-Brain-Barrier disruption from disease burden in ALS through Wnt5a/7a" the authors study the timely origin of Blood-Spinal Cord-Barrier (BSCB) disruption in different mouse models of ALS. Hereby, they show that BSCB can be uncoupled from motor neuron degeneration. Instead, BSCB is associated with early excitatory MN dysfunction. Furthermore, they show that BSCB can be restored by targeting D(Gq), but not D(Gi) G-protein coupled signaling in astrocytes. Mechanistically, Wnt5a and Wnt7a secreted by astrocytes are important to maintain the BSCB. The findings of these studies are interesting and novel as they shed light onto the mechanistic role of astrocytes and the endothelium in contribution to ALS phenotype.

Though the results are encouraging and of important impact for the field, there are some limitations of the study that should be addressed before publication.

Major comments:

- The authors use different mouse models of ALS that, according to the literature, present severe or no motor neuron loss. However, the data shown in Figure 1 is not clearly correlated to the state of motor neuron loss or cell death in the respective mouse model. Even though the data presented has been related to already published findings, the authors

should provide stainings of spinal cord motor neurons and/or cell death (TUNEL, cCasp-3) in combination with the Claudin-5/vessel staining to transmit their hypothesis that Claudin-5 breaks happen independently of motor neuron cell death, more clearly.

> *We provide now a full estimate of MN loss in the spinal cord of the four ALS mouse models considered. Comparable spinal cord sections were samples across the lumbar spinal cord for all four lines. We observed a loss of MN at P50 and P80 (but not at P20) in the SOD1^{G93A} line, a loss of MN at P270 and P450 (but not at P150) in the FUS^{ΔNLS/+} line, a non progressive modest loss of MN in the TDP-43^{G298S} line (comparable at P150, P360 and P510); in the Tbk1^{+/-} mice, no decrease in MN number was observed at any of the two timepoints. These findings are in agreement with previously published characterizations of the ALS models. Since cell death is a short-lived event, the chances of capturing a substantial number of apoptotic MN in a cross-sectional assessment was low, and therefore we decided to avoid an apoptotic cell measurement. We have added the MN loss evidence by staining for ChAT⁺ MNs in Figure S2.*

- Throughout the study, Collagen IV has been used as blood vessel marker. The disadvantage of this marker though is that it only stains the endothelial cell basal membrane. It could be that vessel remodeling takes place during ALS and blood vessels regress. In that case, the Collagen IV matrix could still be detected, though there are no blood vessels anymore. This might lead to a false positive assumption of Claudin-5 breaks resulting from the complete absence of blood vessels.

> *We provide now the immunostaining for the CD31/PECAM, constitutively expressed on the luminal and abluminal surfaces of endothelial cells (Caligiuri, 2019; Feng et al., 2004) together with the collagen-IV immunostaining, to demonstrate that >98% (within the limitations of the staining efficiency) of vascular structures identified by collagen-IV contain actual endothelial cells and are not “empty tubes”. Also, it must be noted that the CLN-5 and ZO-1 ribbons are often detectable before and after the so-called “break”, indicating the continuity of the vessel endothelial lining, and that DAPI-stained nuclei are seen all along the collagen-IV lining. We have added the CD31 evidence in Figure S3.*

- In line with the above comment, is it known that only vessel permeability is affected, or is the blood vessel density per se also affected in ALS?

> *We have now added an assessment of vascular density in the grey matter of the ventral horn of lumbar spinal cord samples (systematically sampled across the lumbar spinal cord) in the four ALS mouse models. We have identified a non-progressive modest decrease in vascular density in all four of the ALS models considered. We have added this piece of information and briefly discussed its implications. The data are displayed in Figure 3.*

- To complement the Claudin-5 analysis, the authors should use a functional assay for the BSCB and inject a fluorescent tracer to analyze vessel permeability. This analysis should be done in both basal disease conditions as well as in the rescue experiments targeting astrocytes and MN excitability. Alternative, staining for fibrinogen and/or immunoglobulin would be important to really demonstrate extravasation and therefore compromised barrier function.

>We now provide data showing albumin deposition in the parenchyma of the spinal cord as a functional measure of BSCB impairment. We demonstrate extravasated albumin deposits already at P20 in the mutant SOD1 line, with a strong increase in the albumin deposits observed at later timepoints (P50, P80). We also detected a substantial albumin deposition in the spinal cord of mutant *FUS*^{ΔNLS/+} (P150, P270, P450), *TDP-43*^{G298S} mice (P150, P360 and P510) as well as *Tbk1*^{+/-} (P270, P450). Taken together, these findings show that the structural disruption of the BSCB is associated with an increased permeability of the brain microvasculature at all timepoints considered. Related data are shown in Figure 5.

- Is only the localization affected or also the total protein levels of Claudin-5? Is the mechanism specific for Claudin-5, or are other junctional proteins affected as well (e.g. VE-Cadherin)?

>We now provide evidence based on the immunostaining for the tight-junction zonula occludens-1 that the focal disruption observed in claudin-5 immunolocalization is also detectable for ZO-1, with a similar time course in the four mouse lines. Since the disruption of claudin-5 is focal in nature and affects only vessels in the ventral horn, biochemical analysis of the whole spinal-cord homogenate is unsuited to investigate this disease manifestation due to the substantial dilutional effect. Data displayed in Figure 2

- The efficiency of MN excitation/inhibition after viral delivery of actPSAM/inhPSAM should be demonstrated to show that the model system is indeed working as expected.

> We have exploited the activity-dependent phosphorylation of CREB on Serine 133 (Wu et al., 2001; Moore et al., 1996) and the induction of the immediate-early gene *DREAM* () as proxies of MN firing. We have verified that expression of actPSAM and administration of the effector PSEM³⁰⁸ result in the upregulation of phosphoCREB and *DREAM* in the nucleus of MN, whereas the expression of inhPSAM and the administration of the effector PSEM causes the significant decrease in the levels of phosphoCREB and *DREAM*. Thus, these measurements, albeit relying on activity markers, demonstrate the validity of the chemogenetic strategy we have employed. These findings are in agreement with the recently-reported chemogenetic induction of *c-Fos* in motoneurons (Baczyk et al., 2020). We have added these new data in the Results chapter and in Figure 6.

- The authors claim that increasing Gi signaling in astrocytes also restores later disease stages. However, at least from the data provided, this conclusion is not fully supported. The authors show that the first, mild Claudin-5 breaks occur at P20 in the SOD mutant, whereas the peak of BSCB starts at around P50. Instead of restoring Gi signaling between P30-P50, the authors should treat the mice at later disease stages (e.g. P50) and analyze the mice at P80.

>According to the recommendations of the Editor, we have not addressed this point at experimental level. We have nevertheless discussed the implications and the possible limitations of the present dataset.

- Is astrocytic endfeet coverage affected in the different models of ALS used?

>We now provide a full quantification of AQP+ vessel coverage for all four ALS models across all timepoints. AQP+ coverage is decreased already at the earliest stages of disease (before MN loss in SOD1 and FUS^{ΔNLS/+} mice, and at the earliest timepoint tested in TDP-43^{G298S} and Tbk1^{+/-} mice) but remains stable across timepoints; these data are now added as new Figure 4. The decreased AQP4+ coverage is in line with the disturbed CLN-5 and ZO-1 distribution and with the increased albumin extravasation, all pointing toward early and substantial disruption of the BSCB.

- How can the authors relate the restoration of Claudin 5 that occurs with the astrocytic Gi/Gq with the recovery of ALS symptoms? It could be that "protection of the BSCB" is a parallel event and not astrocytic dependent. It could be that astrocytes are also signaling to neurons, microglia etc, which could then lead to the effect on neurons. It is recommended to further interpret their data with other potential mechanisms, in the discussion of the paper.

>We have now clarified (in the Discussion) the evidence in favour of a direct astrocyte-endothelial cells interactions while stressing the potential alternative interpretations. It is possible that i) D(Gi) alters Wnt5a and Wnt7a expression in astrocytes but the effect on the BSCB is due to alternative mediators: it is for sure possible that additional mediators may be involved, however the role of Wnt5a/7a in inducing a tight BSCB has been clearly and repeatedly established (Stenman et al., 2008; Liebner et al., 2008; Tran et al., 2016; LeBlanc et al., 2019; Wang et al., 2018; Cho et al., 2010; Laskatorini et al., 2019; Betz et al., 2019; Artus et al., 2004), making a direct connection between Wnt5a/7a upregulation and restoration of BSCB highly possible (also considering that the administration of the Porcupine inhibitor alone is sufficient to further impair the BSCB); ii) D(Gi) or D(Gq) activation in astrocytes may have indirect effect through neurons (since both may induce the release of gliotransmitters; Durkee et al., 2019): this point was directly addressed by the multiplexed chemogenetic experiment in which motoneuronal firing was reduced by inhPSAM while D(Gi) or D(Gq) were activated in astrocytes; whereas the effects of D(Gq) were largely abolished by the inactivation of motoneuron firing (thus suggesting that BSCB effects of astrocytic D(Gq) may be indirect, the effects of D(Gi) were not modified by concomitant MN inactivation, ruling out an indirect effect through MN excitation. Furthermore, astrocytic D(Gi) effects are no longer seen upon activation of MN firing by actPSAM, suggesting that the two effects lie in the same pathway. iii) the upregulation of Wnt5a/7a induced by D(Gi) may affect other non-neuronal cells (e.g., microglia) and then indirectly affect the integrity of BSCB. Although this model is theoretically possible and worth of further investigation, the direct effects of Wnt5a/7a on BSCB integrity are well established (Stenman et al., 2008; Liebner et al., 2008; Tran et al., 2016; LeBlanc et al., 2019; Wang et al., 2018; Cho et al., 2010; Laskatorini et al., 2019; Betz et al., 2019; Artus et al., 2004). Microglia does respond to astrocytic Wnt (Ouali Alami et al., 2018) but Wnt signaling is thought to cause microglial proliferation and substantial pro-inflammatory activation (Halleskog et al., 2011; 2012), although the ultimate outcome may depend on the context (Halleskog et al., 2013). Thus, indirect actions of astrocytic D(Gi), if any, remain to be further investigated.

Minor comments:

- The authors study different disease time points defined as 'presymptomatic, early symptomatic, fully symptomatic stages'. Experimental details should be provided on how these categories have been defined.

>We have now clarified the characterization of the timepoints considered for each line based on the appearance behavioural or histological signs of the disease. For the well-understood high-copy SOD1^{G93A} line, at P20 no biochemical or histological abnormality is known besides the appearance of misfolded SOD1 (Saxena et al., 2009; Saxena et al., 2013); the first wave of neuromuscular junctions denervation appears at P50-55 (Pun et al., 2006) and detection of neurological impairment and body weight loss takes place at about P80 (Ouali Alami et al., 2018; Boillee et al., 2006). Thus, the three timepoints correspond, respectively, to a stage with no histological or biochemical abnormality, a stage corresponding to the first wave of denervation and a stage corresponding to the onset of neurological disability. For the FUS^{ΔNLS/+} mice, statistically-significant abnormalities in motor behaviour appear in some (but not all) tests at the age of approx. 9-10 months (Scekic-Zahirovic et al., 2017), with progressive worsening after that point (up to the age of approx 20 months). We selected one timepoint corresponding to the age of appearance of the first neurological signs (P270, approx 9 months of age), one timepoint corresponding to a stage in which no behavioural impairment is detectable (Wiesner et al., 2018; Scekic-Zahirovic et al., 2017) and a timepoint substantially later in disease progression (P450); logistical issues prevented the follow-up to 20 months of age.

For the transgenic TDP-43^{G298S} line, abnormalities in motor behaviour are detected as soon as the first timepoint testable (Wiesner et al.2018, supplementary figure 4) and remain stable over time in some domains (grip strength, inverted grid test) whereas decline further after P150 in the rotarod performance. Therefore, one timepoint was selected at P150 and two more timepoints were selected at P360 (long after the plateau in the performance decline in the rotarod) and one even later at P510.

For the Tbk1^{+/-} line, no clinical landmark can be used to establish the disease stage since these mice do not display any neurological impairment (Brenner et al., 2019). We considered two timepoints (P270, P450) which were comparable to the ones used for the FUS^{ΔNLS/+} and TDP-43^{G298S} lines.

We have now added the detailed explanation for the selection of the timepoints to the result paragraph.

- It is sometimes difficult to follow the reasoning behind certain experiments, especially the part about G-protein coupled signaling pathways in astrocytes. Please provide a better introduction of the scientific background. Also, many abbreviations, such as 'Gi, Gs or Gq' are not introduced, making it difficult to read the abstract and introduction.

>We have expanded the background introduction and clarified the rationale of the chemogenetic experiments performed on astrocytes: "The PSAM/PSEM experiment established a causal link between MN activity and integrity of the BSCB in ALS. Two sets of consideration let us hypothesize that astrocytes may provide a link between neuronal firing and BSCB in this condition: i) astrocytes sense and respond to increased synaptic activity and neuronal firing by detecting the release (among others) of ATP, glutamate and acetylcholine (Kofuji and Araque, 2020; Durkee and Araque, 2019) as well as K⁺ ions (Simon and Needergard, 2004); ii) astrocytes are involved in the control of BSCB differentiation and integrity (Abbott et al., 2006; Liebner et al., 2018) by releasing GDNF

(Igarashi et al., 1999), angiopoietin-1 and bFGF (Lee et al., 2003). Since neuronal activity is largely sensed by astrocytes through different families GPCR signaling coupled to Gi, Gs or Gq subgroups of G-proteins (Kofuji and Araque, 2020), we speculated that controlling Gi, Gs or Gq signaling through designer receptors (DREADDS, coupled to either Gs [DGs], Gi [D(Gi)] or Gq (D(Gq))) we could evoke astrocytes responses related to increased neuronal activity and possibly restore the integrity of the BSCB without intervening on MN themselves”

Abbreviations regarding G proteins are standard nomenclature (as used in scientific literature and textbooks) and are usually not spelled, since the original name of cAMP Stimulatory GTP binding protein (Gs) or cAMP Inhibitory GTP binding protein have now only historical significance and do not reflect the complex biochemistry of these signal-transducing proteins.

- The manuscript should be edited/revised by a professional/native speaker due to some minor, but repetitive language mistakes.

> *The manuscript has now been edited by a native English speaker.*

- Please provide high resolution figures as well as better representative images.

> *We have now replaced or enhanced most of the figures through the paper.*

- Figure 2 could be nicely complemented with a graph showing infection efficiency of the MN population.

> *All figures depicting chemogenetic experiments, including Figure 7 (formerly figure 2) now show in panel A an overview of the spinal cord displaying 1) the efficiency of MN infection (>90%) and 2) the selectivity of the injection, with <5% infected MN in the contralateral side.*

- Show representative images of LC3A staining (autophagy marker) as complementary to the presented quantification graph (Figure 4).

> *Representative images have been now added to Figure 10 (formerly Figure 4).*

- In Figure EV1 the authors show astrocytic endfeet coverage, however they do not use AQP4 as a marker for the endfeet.

> *We have now added the immunostaining for AQP4 not only for (currently) Figure 9 but we also provide a full timecourse of the vessel coverage by AQP4⁺ processes for the 4 ALS mouse lines across the different timepoints (Figure 4).*

- In Figure EV2 there is no GFAP/AQP4 staining showing astrocytes in SOD1 mice. Do the authors see a loss in the number of astrocytes or just activity alterations in this model?

> *In addition to the new dataset on perivascular AQP4 expression in the four ALS lines (Figure 4), now we also provide in Figure S4 also the quantification of the level of astrogliosis (measured either as number of astrocytes per area unit (a measure of astrocyte proliferation) either as GFAP⁺ area per area unit (a measure of astrocyte hypertrophy)). We*

observe a progressive increase in astrogliosis in the ventral horn of the spinal cord of SOD1^{G93A} mice, as previously reported (e.g., Ouali Alami et al., 2018). Moreover, we observe a degree of astrogliosis in FUS^{ΔNLS/+}, TDP-43^{G298S} and Tbk1^{+/-} mice as well. Although reactive gliosis is not surprising in the context of neurodegeneration (such as the loss of some MN contingent, as demonstrated by the MN counts that we now present in Figure S2), gliosis takes place in the context of reduced AQP4 coverage of vessels.

- Figure 6: Wnt5a/Wnt7a staining/ISH is not visible in the images they show. Also, based on what they describe, Wnt staining should also be detected in MNs. Would combine Wnt ISH with MN specific staining to further demonstrate this. Maybe also perform western blot to show the expression.

> We have now replaced the set of representative images depicting single-molecule mRNA in situ hybridization for Wnt5a and Wnt7a with a new one with higher contrast and better color code to improve visualization. Wnt5a and Wnt7a are expressed in multiple cell types in the brain and spinal cord, with multiple functions. Since this paper focuses on astrocytic Wnt5a/7a expression in relationship to their role in regulating the BSCB integrity in ALS, we have opted not to proceed with the analysis of Wnt expression in MN, since this effect, albeit interesting, would be tangential to the current focus. Likewise, WB analysis has not been performed since this would sample Wnt5a/7a in homogenates of the whole spinal cord, thus including Wnt proteins produced by several different cell types at once.

- Figure4- Insets to see the misfolded SOD1-the image quality is very poor. What do the grey cells indicate-which marker is that?

> In current Figure 10, the grey channel is used to highlight the different disease markers (mis SOD1, p62, KDEL, LC3A) whereas the identity of the cells (motoneurons) is provided by the VACHT staining (appearing as bright C-boutons surrounding the motoneurons and as fainter cytoplasmic staining). We have replaced the set of representative images to improve the visibility.

- What is the molecular mechanism of suppressed Wnt5a and 7a release due to MN dysfunction? It would be nice if the authors could further elaborate that point in the discussion

> To the best of our knowledge, nothing is known about the molecular mechanisms linking astrocytic Wnt5a/7a expression to MN signals and very little is known about the regulation of astrocytic Wnt5a/7a in adults. At this stage, we can only speculate that GPCR-mediated signals may be relevant to this process, based on the abundant expression in astrocytes of GPCRs involved in synaptic/neuronal activity monitoring (Kofuji and Araque, 2020); furthermore, astrocytic NF-κB may be also involved, based on the conditional over-activation experiments we have previously reported (Ouali Alami et al., 2018). We have added a note to the discussion on this issue.

Reviewer #3 (Comments to the Authors (Required)):

The manuscript by Ouali Alami et al. provides interesting data describing the relationship of motoneuron (MN) activity and blood-spinal Cord-Barrier (BSCB) under the pathological conditions in a mouse model of Amyotrophic Lateral Sclerosis (ALS). Specifically, the authors show that BSCB impairment was independent of MN loss in different models of ALS, namely in SOD1(G93A), FUS(Δ NLS), TDP43(G298S) and TBK1^{+/-} ALS mouse models. Interestingly, in SOD1G93A ALS mice the BSCB was restored by increased MN firing, but worsened by MN inactivation, using a chemogenetic approach. The authors could further demonstrate that specific modulation of GPCR signaling in astrocytes (ACs) by clozapine-N-oxide (CNO)-inducible activation of D(Gs), D(Gi) and D(Gq) signaling downstream of the GFAP promoter after AAV infection.

The authors show that D(Gi) and D(Gq) were able to restore BSCP leakage in SOD1(G93A) mice, whereas D(Gs) had no significant effect. The BSCP restoration had no beneficial effect on MN disease markers, which was interpreted by the authors to be indicative of independent processes. Finally, Alami et al. provide evidence for a decreased expression of Wnt7a and Wnt5a in SOD1(G93A), which however could strongly be enhanced by Gi signaling, but further decreased by MN inactivation.

Together, the presented data this manuscript demonstrate that in ALS the BSCB is indeed impaired in various mouse models of the disease, and that this is related to MN firing as well as AC GPCR signaling.

The data is solid and the presentation of the data is very good. However, to augment the scientific merit of the manuscript, some issues require the authors attention:

Major points

One major concern of the reviewer regards the Wnt pathway activity in the vasculature of the spinal cord. The authors assume that the down-regulation of Wnt5a and Wnt7a in ACs leads to reduced Wnt activation in endothelial cells (ECs). However, this has to be supported by experimental evidence. The authors should stain for Lef1 and/or isolate vessels to monitor expression of Wnt target genes like Axin2, Nkd1 or Sox17 in ECs. Wnt5a might also inhibit the beta-catenin-driven Wnt pathway. The authors should take this also into consideration.

Does DREADD activation that leads to Wnt5a and Wnt7a expression also augment Wnt/beta-catenin signaling in ECs?

> Following the recommendation of the editor, we have addressed these points in the Discussion :“Upon increased availability of Wnt5a and Wnt7a, it is believed that the activation of the β -catenin signaling cascade in endothelial cells is a key event in inducing and maintaining BSCB specializations. In fact, canonical Wnt signaling through β -catenin is sufficient to induce a BBB-like phenotype in cultured endothelial cells (Laskatorini et al., 2019) and it is critically involved in establishing the BBB phenotype in vivo (Liebner et al., 2008; Chang et al., 2017; Wang et al., 2018; Cho et al., 2017) whereas reduced wnt/beta-catenin signaling spatially corresponds to sites of high BBB permeability in the brain (Benz et al., 2019). Moreover, non-canonical Wnt signaling may contribute to the establishment of the BBB phenotype (Pinzon-Daza et al., 2014). Sox17 has been identified as a major target of the β -catenin pathway in endothelial cells (Corada et al., 2019), together with Lef1 and Ets1 (Roudnicky et al., 2020).”

Additional points are raised below:

Figure 1B, C, D: Vessels of the considerably longer aged (P150-P270) WT mice of the FUS(Δ NLS), TDP43(G298S) and TBK1^{+/-} ALS mouse models also show stretches devoid of

Cldn5. Is this an ageing phenotype that the authors did not observe in the younger (P20) SOD1(G93A) mouse model?

>The age-dependent mild impairment of the Blood-Brain-Barrier is a well-described phenomenon, identified in mice (Stamatovic et al., 2019) as well as in humans (Verheggen et al., 2020). We have now briefly addressed this point.

Figure 6 C-N: The staining and hence the quantification of Wnt5a and Wnt7a is not convincing. Other, quantitative methods such as western blotting and qRT-PCR would be ideal to complement the finding.

> WB or qPCR deliver a result based on the average expression within the sampled tissue and therefore would dilute any effect seen in ventral horn astrocytes with the rest of the spinal cord tissue; therefore, they are unfortunately unsuitable to confirm the results obtained with single-molecule in situ mRNA hybridization. The effects we have reported are highly local and therefore we believe are best characterized by high sensitivity techniques that preserve the "spatial component" of the observed effects. Presented in Figure 6 C-N are single-molecule mRNA hybridization (not immunostaining) in which each fluorescent dot represents a single mRNA molecule, hence enabling a digital quantification of mRNA expression. Because of the comparatively small volume of astrocytic cytoplasm, the number of mRNA molecules at baseline is in the 5-20 per cell. We have now prepared an enhanced set of representative images to highlight the biological effects at single cell and single mRNA molecule resolution, and we have stressed in the text the type of technique used.

Figure EV1 and EV2: The staining of GFAP, and in particular of Aqp4, do not convincingly show vessel coverage, leaving doubt with regard to the quantification. Please provide improved images.

>We have now provided a full quantification of perivascular AQP4 levels at every timepoint for each ALS mouse line and we have provided a full set of new images in Figure 4; furthermore, we have replaced the previous AQP4 panels in Supplementary figure 5. We have integrated the quantification of AQP4 with the quantification of global astrocytosis, shown in Figure 9.

Minor points

Page 6, line 9: Which software has been used for vessel and leakage quantification?

> We used the ImageJ image analysis software for vessel quantification. This information is now added to the Methods section.

Page 17, line 19: Please change "..., and it sufficient to induce a BBB-like phenotype in endothelial cells [4] and on Wnt7a, a critical mediator of the establishment and maintenance of the BBB phenotype [54, 12]." into "..., and is sufficient to induce a BBB-like phenotype in endothelial cells [4] and on Wnt7a, a critical mediator of the establishment and maintenance of the BBB phenotype [54, 12]."

> We thank the reviewer for the advice. We have amended the text as suggested.

Overall the manuscript by Alami et al. describes interesting findings. Hence the reviewer recommends the manuscript for publication after major revision.

August 14, 2020

RE: Life Science Alliance Manuscript #LSA-2019-00571-TR

Prof. Francesco Roselli
Ulm University
Neurology
Helmholtzstrasse 8/2
Ulm 89081
Germany

Dear Dr. Roselli,

Thank you for submitting your revised manuscript entitled "Multiplexed chemogenetics in astrocytes and motoneurons restore Blood-Spinal Cord-Barrier in ALS". We would be happy to publish your paper in Life Science Alliance pending final revisions necessary to meet our formatting guidelines.

Please make the following edits as you prepare the manuscript for publication at Life Science Alliance,

- please add an ORCID ID for the secondary corresponding author--they should have received instructions on how to do so
- please add a callout for Figures 4G-H; Fig. 11D,F; and Fig. S4C in the main manuscript text
- please double check your figure callouts (you have a callout for Figure S6, but there isn't a figure S6)
- please add the supplementary figure legends to the main manuscript text
- please use the [10 author names, et al.] format in your references (i.e. limit the author names to the first 10)
- please provide scale bars for figures 7B, 8B, 11B and 13H
- please clarify that the higher mag images in Figure 4 A, C, E, G do not include DAPI staining, in the Fig 4 legend
- please make sure that the panel labels in Figure S3 match with the legend of supplemental fig 3.

A. FINAL FILES:

B. MANUSCRIPT ORGANIZATION AND FORMATTING:

Sincerely,

Shachi Bhatt

Executive Editor
Life Science Alliance

Reviewer #1 (Comments to the Authors (Required)):

The authors have extensively reviewed the manuscript and done this to my satisfaction. I appreciate that a biochemical analysis (points 2 and 9) is not possible and that human tissue is scarce (point 7).

August 31, 2020

RE: Life Science Alliance Manuscript #LSA-2019-00571-TRR

Prof. Francesco Roselli
Ulm University
Neurology
Helmholtzstrasse 8/2
Ulm 89081
Germany

Dear Dr. Roselli,

Thank you for submitting your Research Article entitled "Multiplexed chemogenetics in astrocytes and motoneurons restore Blood-Spinal Cord-Barrier in ALS". It is a pleasure to let you know that your manuscript is now accepted for publication in Life Science Alliance.

While preparing the manuscript for production, we noticed that the supplemental figure 3 legends do not match with the panel labels in S3 figure (legends go as A, C, E, G; while the S3 figure panels are labeled A, B, C, D). We have asked our copy editors to correct this before the proofs stage. Please expect this edit in your manuscript proofs, and let us know if you disagree with the edits.

DISTRIBUTION OF MATERIALS:

Congratulations this interesting study. I hope you found the review process to be constructive and are pleased with how the manuscript was handled editorially. We look forward to future exciting submissions from your lab.

Sincerely,

Shachi Bhatt, Ph.D.
Executive Editor
Life Science Alliance